# *Arabidopsis* TRM5 encodes a nuclear-localised bifunctional tRNA guanine and inosine-N1-methyltransferase that is important for growth

Qianqian Guo[1‡], Pei Qin Ng[2‡], Shanshan Shi[1], Diwen Fan[1], Jun Li[2], Jing Zhao[2], Hua Wang[3], Rakesh David[4], Parul Mittal[5], Trung Do[2], Ralph Bock[6], Ming Zhao[1], Wenbin Zhou[1]*, Iain Searle[2]*

1 Institute of Crop Sciences, Chinese Academy of Agricultural Sciences, Beijing, China, 2 School of Biological Sciences, School of Agriculture, Food and Wine, The University of Adelaide and Shanghai Jiao Tong University Joint International Centre for Agriculture and Health, The University of Adelaide, Adelaide, Adelaide, Australia, 3 National Key Laboratory of Plant Molecular Genetics, CAS Center for Excellence in Molecular Plant Sciences, Shanghai Institute of Plant Physiology and Ecology, Chinese Academy of Sciences, Shanghai, China, 4 ARC Centre of Excellence in Plant Energy Biology, School of Agriculture, Food and Wine, University of Adelaide, Adelaide, Australia, 5 Adelaide Proteomics Centre, School of Biological Sciences, The University of Adelaide, SA, Australia, 6 Max Planck Institute of Molecular Plant Physiology, Am Mühlenberg, Potsdam-Golm, Germany

‡ These authors are joint first authors on this work.
* Iain.Searle@adelaide.edu.au(LS); zhouwenbin@caas.cn(WZ)

**Data Availability Statement:** The data used in this paper is submitted into NCBI with the GEO accession number GSE114898. Proteomics raw data has been deposited on iProX, with under

## Abstract

Modified nucleosides in tRNAs are critical for protein translation. $N^1$-methylguanosine-37 and $N^1$-methylinosine-37 in tRNAs, both located at the 3'-adjacent to the anticodon, are formed by Trm5. Here we describe *Arabidopsis thaliana AtTRM5* (At3g56120) as a Trm5 ortholog. *Attrm5* mutant plants have overall slower growth as observed by slower leaf initiation rate, delayed flowering and reduced primary root length. In *Attrm5* mutants, mRNAs of flowering time genes are less abundant and correlated with delayed flowering. We show that *AtTRM5* complements the yeast *trm5* mutant, and *in vitro* methylates tRNA guanosine-37 to produce $N^1$-methylguanosine ($m^1G$). We also show *in vitro* that AtTRM5 methylates tRNA inosine-37 to produce $N^1$-methylinosine ($m^1I$) and in *Attrm5* mutant plants, we show a reduction of both $N^1$-methylguanosine and $N^1$-methylinosine. We also show that AtTRM5 is localized to the nucleus in plant cells. Proteomics data showed that photosynthetic protein abundance is affected in *Attrm5* mutant plants. Finally, we show tRNA-Ala aminoacylation is not affected in *Attrm5* mutants. However the abundance of tRNA-Ala and tRNA-Asp 5' half cleavage products are deduced. Our findings highlight the bifunctionality of AtTRM5 and the importance of the post-transcriptional tRNA modifications $m^1G$ and $m^1I$ at tRNA position 37 in general plant growth and development.

accession number IPX0001222000 and is available at https://www.iprox.org//page/project.html?id=IPX0001222000.

**Funding:** The research was partially supported by ARC grant FT130100525, partially funded an Australia-China Science and Research Fund grant ACSRF48187 awarded to I.S. and an APA awarded to PQ.N and J.L. This research was also partially supported by the National Natural Science Foundation (31570234) awarded to W.Z., W.Z. was supported by the Innovation Program of Chinese Academy of Agricultural Sciences and the Elite Youth Program of the Chinese Academy of Agricultural Science.

**Competing interests:** The authors have declared that no competing interests exist.

# Introduction

RNA has over 100 different post-transcriptional modifications that have been identified in organisms across all three domains of life [1–5]. While several RNA modifications have been recently identified on mRNAs in yeast, plants, and animals, tRNAs are still thought to be the most extensively modified cellular RNAs [6–9]. These tRNA modifications are introduced at the post-transcriptional level by specific enzymes. These enzymes recognize polynucleotide substrates and modify individual nucleotide residues at highly specific sites. Some tRNA modifications have been shown to have a clear biological and molecular function [10, 11]. Several tRNA modifications around the anticodon have been demonstrated to have crucial functions in translation, for example, by enhancing decoding [12], influencing the propensity to ribosomal frameshifting or facilitating wobbling [13–15]. Modifications distal to the tRNA anticodon loop can also directly influence the tRNA recognition and/or translation process [16] or can have roles in tRNA folding and stability [1, 17]. However, the precise functions of many tRNA modifications still remain unknown despite often being conserved across species. Often, loss of a tRNA modification does not negatively impair cell growth or cell viability under standard laboratory growth conditions [18]. However, under environmental stress, such mutants display a discernible phenotype [11].

The tRNA anticodon loop position 37 is important to maintain translational fidelity, prevent frameshift errors and translational efficiency [10, 19–21], and almost all tRNAs are modified at this site. There are two prominent modifications at tRNA position 37: $N^1$-methyl-guanosine ($m^1G37$) and 1-methylinosine ($m^1I$). Trm5 in humans, yeast, and *Pyrococcus abyssi* has been described as having multifunctionality [22–24]. $N^1$-methylation of guanosine at position 37, $m^1G37$, is performed by TrmD-type enzymes in bacteria, functionally and evolutionarily unrelated Trm5-type proteins in Archaea and Eukaryote, Trm5p in yeast *Saccharomyces cerevisiae*, and TRMT5 *in vivo* in humans [21–23, 25–27]. Trm5p complete loss of function mutants in yeast *Saccharomyces cerevisiae* are lethal whereas mutations in TRMT5 lead to multiple respiratory-chain deficiencies and a reduction in mitochondrial tRNA $m^1G37$ [10, 23, 26]. In humans, TRMT5 (tRNA methyltransferase 5) catalyses the formation of $m^1G37$ *in vivo* on mitochondrial tRNA$^{Pro}$ and tRNA$^{Leu}$ [22, 28]. $N^1$-methylguanosine has been described in eukaryotic tRNAs at two positions; at position 37 catalysed by Trm5, and the other at position 9 catalysed by Trm10 [29]. In contrast to bacteria TrmD, which requires a guanosine at position 37, human TRMT5 can also recognise and methylate inosine at position 37 with some limited activity [22]. Similarly,Trm5p has also been shown to catalyse inosine to $m^1I$ modification in yeast in a two-step reaction, where the first adenosine-to-inosine modification was mediated by Tad1p [11, 18, 26, 30]. As $m^1G$ is an intermediate during the modification of guanosine to wybotusine (yW), tRNAs from *trm5* mutants were also devoid of yW [28]. The yeast Trm5p protein has been shown to be localised to the cytoplasm and mitochondria and it is thought that Trm5p protein present in the mitochondria is required to prevent unmodified tRNA affecting translational frameshifting [23, 26]. In the unicellular parasite *Trypanosome brucei*, Trm5 was located in both the nucleus and mitochondria and reducing Trm5 expression led to reduced mitochondria biogenesis and impaired growth [31]. Interestingly, Trm5 and $m^1G37$ were shown to be essential for mitochondrial protein synthesis but not cytosolic translation [31].

Little is known about tRNA modifying enzymes in plants, especially the plant homolog of this bifunctional methyltransferase. Here, we report the identification and functional analysis of *AtTRM5* (At3g56120) from the model plant *Arabidopsis thaliana*. We demonstrate that *Attrm5* mutant plants are slower growing, have reduced shoot and root biomass and display late flowering. Furthermore, we demonstrate that *in vitro TRM5* is required for $m^1G37$ and

$m^1I37$ methylation at the position 3' to the anticodon and *in vivo* tRNAs enriched from *Attrm5* plants have reduced $m^1G$ and $m^1I$.

## Results

### Identification of At3g56120 as a TRM5 homolog

In yeast (*Saccharomyces cerevisiae*), $m^1G37$ nucleoside modification is catalysed by Trm5p/ScTrm5 [26]. We searched for *Arabidopsis thaliana* homologs by using blastp and HMMER and identified a high confidence candidate, At3g56120, with 49% similarity to ScTrm5 (S2 Fig). Alignment of At3g56120 with yeast, human, *Drosophilia*, *Pyrococcus*, and *Methanococcus* Trm5 homologs identified three conserved motifs and catalytically required amino acids (R166, D192, E206) present in At3g56120 (S2 Fig). We subsequently will refer to At3g56120 as AtTRM5. In the *Arabidopsis* genome, AtTrm5 has homology to At4g27340 and At4g04670 and both genes were recently named as TRM5B and TRM5C, respectively (S2 Fig and [32]). We have also identified TRM5 homologs in algae, bryophytes and vascular plants (Fig 1A and S1 Fig). We focussed our experiments on *Arabidopsis* At3g56120/AtTRM5 as the protein had the highest amino acid similarity to yeast ScTrm5.

To functionally characterize AtTRM5 we isolated two T-DNA insertions, SALK_022617 and SALK_032376, and identified homozygous mutant plants for each insertion (Fig 1B). SALK_022617 and SALK_032376 were named *trm5-1* and *trm5-2*, respectively. Next, we measured AtTRM5 mRNA abundance in both mutants and detected almost no transcripts in both mutants (Fig 1C). We generated a genomic construct of AtTRM5 that contained the endogenous promoter, coding region, and UTRs, transformed the construct into *trm5-2* and demonstrated that the AtTRM5 mRNA levels were similar in two complemented lines when compared to wild type plants. Subsequently, the extracted tRNAs from wild type and the *trm5* mutants were purified, digested and modified nucleosides measured by mass spectrometry (Fig 1D). In both *trm5* mutant alleles, nucleoside $m^1G$ levels were reduced to about 30% of the wild type and $m^1G$ levels were restored to wild type levels in both complemented lines (Fig 1D). Nucleoside $m^1G$ is present at tRNA positions 9 and 37 [26], therefore the residual $m^1G$ levels in *trm5* mutants may be the result of tRNA $m^1G$ at position 9. In saying this, we cannot exclude an alternative explanation that the residual $m^1G$ levels in *trm5* plants are the result of activity of either TRM5B or TRM5C.

In yeast, Trm5 has also been reported to also catalyse $m^1I$ on tRNAs [11, 26]. We therefore measured $m^1I$ levels in purified tRNAs from both *Arabidopsis trm5* mutants and wild type control plants. In both *trm5-1* and *trm5-2* mutant alleles, nucleoside $m^1I$ levels were reduced to about 10% of wild type levels and were restored to wild type levels in plants of both complemented lines (Fig 1D). It is possible that the residual $m^1I$ in *trm5* plants may be $m^1I$ at position 57 [4, 33, 34]. In summary, we identified At3g56120 as a TRM5 homolog in *Arabidopsis thaliana*, identified two AtTRM5 mutant alleles, *trm5-1* and *trm5-2*, and both mutants showed a significant reduction in $m^1G$ and $m^1I$.

### AtTRM5 is involved in leaf and root development and flowering time regulation

Before undertaking growth measurements, we grew wild-type Columbia, *trm5-1*, two complemented lines, and two overexpression lines together under long-day conditions, harvested and dried the seeds to minimise any maternal or environmental effects. To observe the early growth stages of seedlings, we grew the six lines (wild type, *trm5-1*, two complemented lines and two overexpression lines) on plates for 10 days (Fig 2A). The *trm5-1* seedlings were

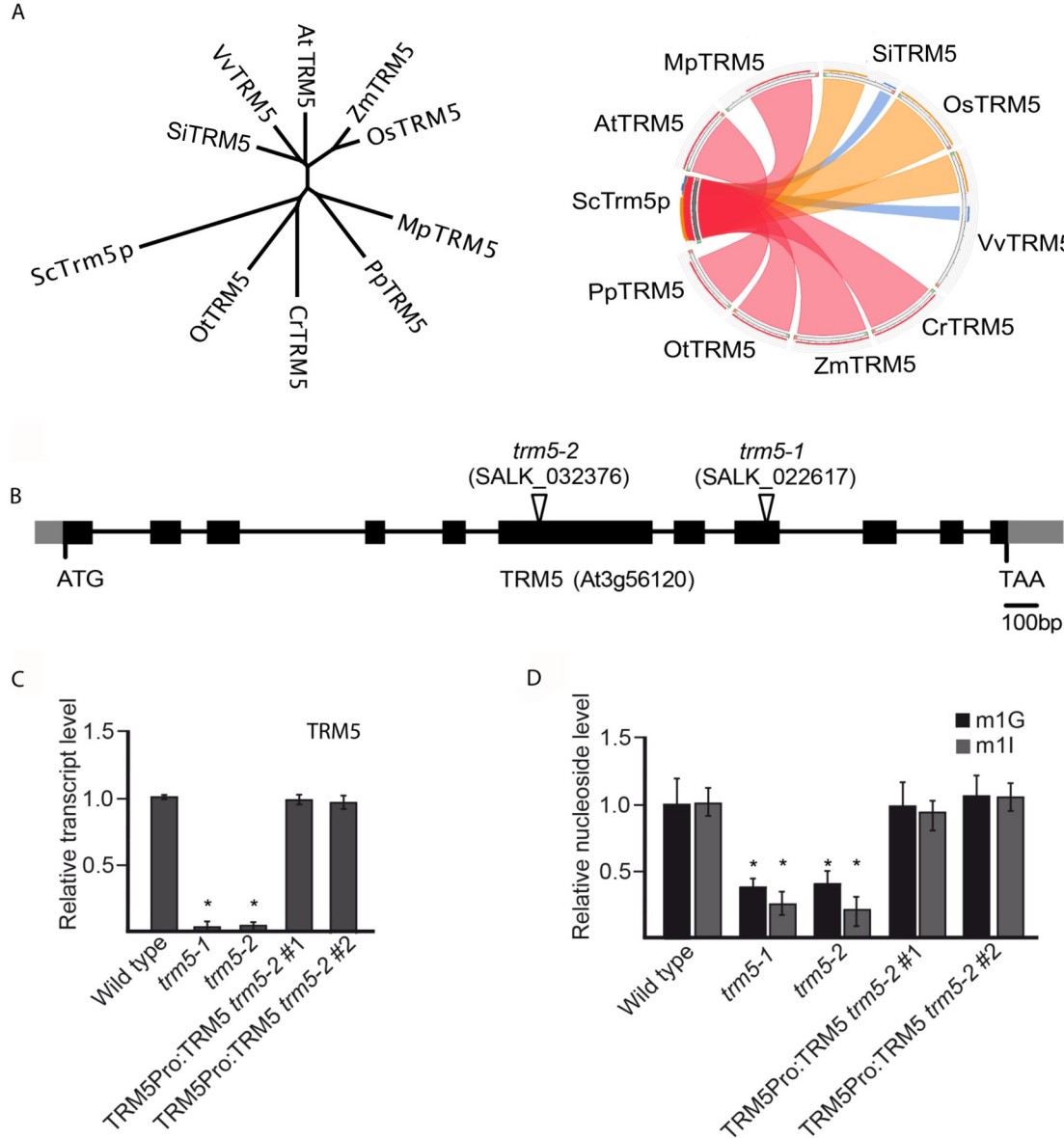

**Fig 1. TRM5 is conserved in plants and has dual-functionality in modifying RNA bases.** (**A**) Unrooted phylogenetic tree and sequence conservation Circos plot of putative TRM5 proteins from yeast (Sc), tomato (Sl), grape (Vv), *Arabidopsis* (At), maize (Zm), rice (Os), *Marchantia* (Mp), *Physcomitrella* (Pp), *Chlamydomonas* (Cr), and *Ostreococcus* (Ot). The ribbons were coloured based on sequence identity, with blue < = 25%, green 25–50%, orange 51–75% and red for 76–99%. (**B**) Exon-intron structure of the putative TRM5 locus (At3g56120) showing the T-DNA insertion sites of the *trm5-1* and *trm5-2* alleles (as indicated by the open triangles). Black boxes and grey boxes represent coding regions and untranslated regions, respectively. (**C**) Relative transcript level detected by qPCR in wild type, *trm5-1* or *trm5-2* seedlings. (**D**) Relative nucleoside level of modification $m^1G$ and $m^1I$ detected by HPLC/MS in wild type, *trm5-1* or *trm5-2* seedlings.

noticeably smaller than the wild type. In contrast, no clear differences were evident between wild type, the complemented and overexpression lines. To rule out the possibility that the reduced growth in *trm5-1* seedlings was due to slower germination, we measured the germination of *trm5-1* and wild type and no difference was observed (S3 Fig). Reduced growth of *trm5-1* roots was also evident on plate-grown plants (Fig 2B). Interestingly, *trm5-1* primary, lateral and total (primary + lateral) root lengths were reduced in *trm5-1* when compared to

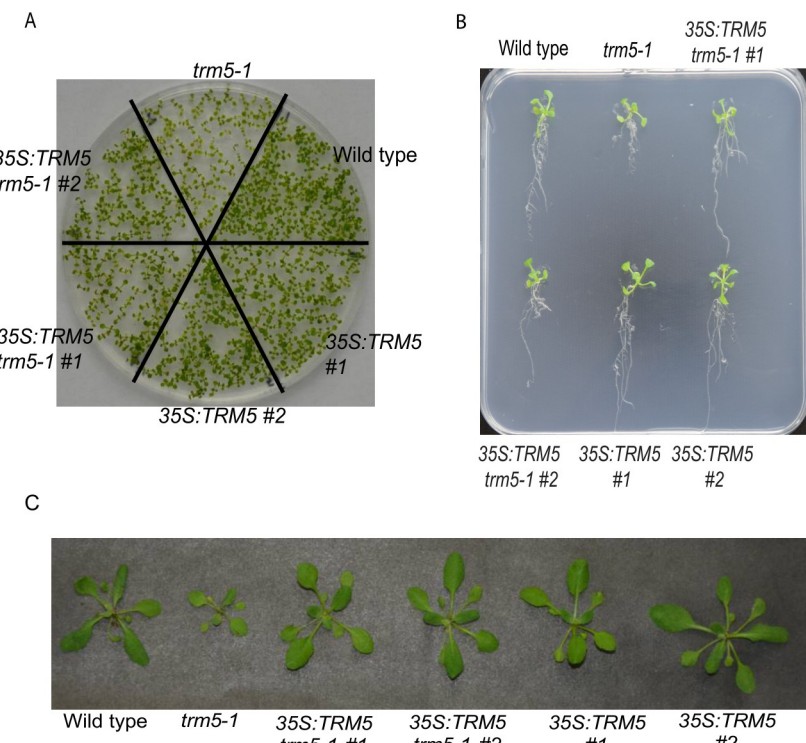

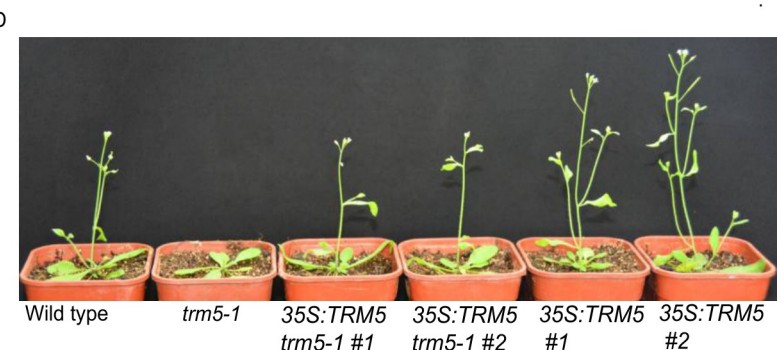

**Fig 2. Phenotype analysis of *trm5*, complemented lines (35S:TRM5 *trm5-1*) and TRM5 overexpression lines (35S:TRM5).** (**A**) Seedlings were sown on ½ MS media plates and grown for 7 days and photographed. (**B**) Seedlings of wild type, *trm5*, complemented lines (35S:TRM5 *trm5-1*), TRM5 overexpression lines (35S:TRM5) were vertically grown on ½ MS medium for 10 days and then photographed. (**C**) Plants were grown on soil under long day photoperiods and photographed 15 days after germination. (**D**) Wild type, *trm5-1*, two complementing (35S:TRM5 *trm5-1*) and two overexpressing lines (35S:TRM5) were grown under long days and representative plants photographed at flowering.

wild-type plants (S4 Fig). We also measured the lateral root number and found that *trm5-1* plants had reduced numbers when compared to the wild type (S4 Fig). In contrast, no differences in the root growth were evident upon comparison of the wild type and the complemented lines. In *TRM5* overexpression lines, primary and lateral root lengths were slightly longer than in the wild type (Fig 2B and S4 Fig).

At inflorescence emergence in wild-type plants grown under long days, we observed reduced rosette leaf numbers, smaller leaves and reduced fresh weight in *trm5-1* plants (Fig 2C and 2D; S3 Fig). Sectioning of the shoot apical meristems of wild type and *trm5-1* plants at wild-type floral transition, confirmed that *trm5* plants were later flowering (S3 Fig). We

measured the flowering time of wild type, *trm5-1*, complemented, and overexpression lines under both short and long days and observed that mutants produced more rosette leaves and flowered later than the wild type (Fig 3A; S3 Fig). Plants overexpressing TRM5 flowered slightly earlier than wild type under both long and short-day conditions (Fig 2D; S3 Fig).

### AtTRM5 is involved in circadian clock and flowering time gene expression

To explore the molecular basis of delayed flowering in *trm5-1*, we measured the mRNA abundance of circadian clock and flowering time related genes by quantitative RT-PCR over a

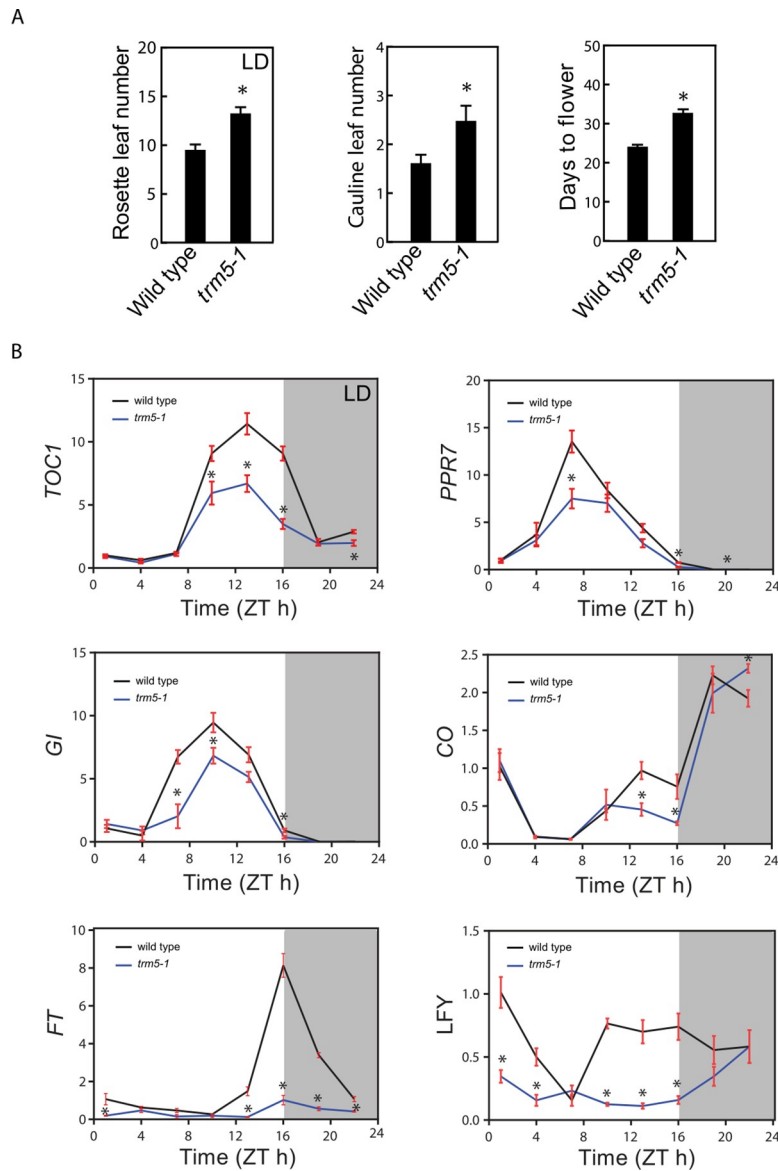

**Fig 3. More leaves, slower flowering and impacted photosynthetic genes in trm5 mutant of *Arabidopsis thaliana*.** (**A**) Rosette leaf number (long-day conditions), cauline leaf number and days to flower (short day conditions) of wild type and *trm5* mutant. (**B**) The mRNA abundance of circadian clock-related genes over a 24-hour period. 17-day-old seedlings of wild type, *trm5-1*, complemented lines (35S:TRM5 *trm5-1*), TRM5 overexpression lines (35S:TRM5) were grown on ½ MS medium for 10 days and then harvested every 3 hours. The expression levels of *TOC1*, *PRR7*, GI, CO, *FT*, and LFY were measured and normalized relative to *EF-1-α*. Data presented are means. Error bars are ± SE, n = 3 biological replicates. An asterisk indicates a statistical difference (*P*<0.05) as determined by Student's t-test. 0 hours is lights on and 16 hours is light off. Shaded area indicates night, whereas Zeitgeber time is abbreviated as ZT.

24-hour period (Fig 3). In *trm5-1* plants, lower abundance of the clock genes *TIMING OF CAB EXPRESSION 1* (*TOC1*), and *PSEUDO RESPONSE REGULATOR 7* (*PPR7*), and the flowering time regulator genes *GIGANTEA (GI), CONSTANS (CO)* and *FLOWERING LOCUS T (FT)* were observed at ZT10 to ZT22. The reduced abundance of the flowering time regulators *GI, CO*, and *FT* in *trm5-1* mutants correlates with delayed flowering. As expected, the downstream floral meristem identity gene *LEAFY (LFY)* had lower abundance at almost all tested time points in the *trm5* mutant when compared to wild type (Fig 3B). The downregulation of circadian-clock related genes were also detected in the RNA-seq data (S2 Table). Together these results support a role for AtTRM5 in plant growth, development, and flowering time regulation.

## AtTRM5 m$^1$G methyltransferase activity

To test AtTRM5 m$^1$G methyltransferase activity *in vivo*, a *Δtrm5* mutant strain in yeast *(Saccharomyces cerevisiae*, Sc) that is defective for the tRNA m$^1$G37 modification was used for genetic complementation. The mutant not only has defective tRNA m$^1$G37 but also a slow growth phenotype when compared to wild type or a congenic strain (Fig 4A). Full-length *ScTrm5* and *AtTRM5* were cloned into yeast expression vectors. From the *AtTRM5* expression vector, a catalytically inactive mutant *Attrm5* R166D was generated by site-directed mutagenesis. After the three vectors had been transformed into the yeast *Δtrm5* mutant, cell growth and m$^1$G nucleoside levels were observed (Fig 4). Not only were the slow growth and nucleoside levels rescued when expressing *ScTrm5* but they were also rescued when expressing *AtTRM5* (Fig 4A and 4B). However, the catalytically inactive Attrm5 R166D did not rescue either the slow growth or m$^1$G nucleoside levels (Fig 4A and 4B).

To test the m$^1$G methyltransferase activity of AtTRM5 *in vitro*, we incubated purified recombinant proteins with tRNA substrates and measured the m$^1$G levels. We expressed *AtTRM5* as a GST fusion protein and purified the recombinant GST-AtTRM5 protein. We also generated a catalytically inactive GST-AtTRM5 recombinant protein by using site directed mutagenesis, expressed and purified the GST-AtTRM5 recombinant fusion protein. In yeast, ScTrm5 methylates tRNA-His-GUG, tRNA-Leu-UAA, tRNA-Asp-GUC and other tRNA isoacceptors [11, 26, 32]. Yeast tRNA-Asp-GUC RNA transcripts were generated *in vitro* by using T7 RNA polymerase, the tRNA transcripts incubated with the recombinant fusion proteins in the presence of AdoMet and m$^1$G nucleoside levels were measured (Fig 4D). m$^1$G was detected only when AtTRM5 was provided (Fig 4D) and in a dosage dependent manner. No m$^1$G was detected when the catalytically inactive mutant AtTRM5 was provided (Fig 4D). To test the specificity of the methyltransferase activity on tRNA-Asp guanine at position 37, the guanine nucleotide was mutated to an adenine nucleotide (tRNA-Asp-A37) and the m$^1$G methyltransferase activity was measured. No m$^1$G was detected after incubation with the fusion proteins (Fig 4D). The overall results of the yeast complementation experiments suggest that guanosine methylation occurred at position 37 of tRNA.

## AtTRM5 tRNA m$^1$I methyltransferase activity

Previously in plants, TAD1 was demonstrated to oxidatively deaminate adenosine at position 37 of tRNA-Ala-(UGC) to inosine, and subsequently methylated by an unknown enzyme to N1-methylinosine (m$^1$I; Fig 5A). Human TRM5 has been reported to methylate tRNA I37 but with limited activity [22]. Given our observation that *Attrm5* mutant plants had reduced m$^1$I (Fig 1D), we asked the question whether AtTRM5 has methyltransferase activity on tRNA I37. We developed a two-step approach, whereby purified AtTAD1 was first incubated with the substrate tRNA-Ala-A37 to produce tRNA-Ala-I37 and then the inosine methyltransferase

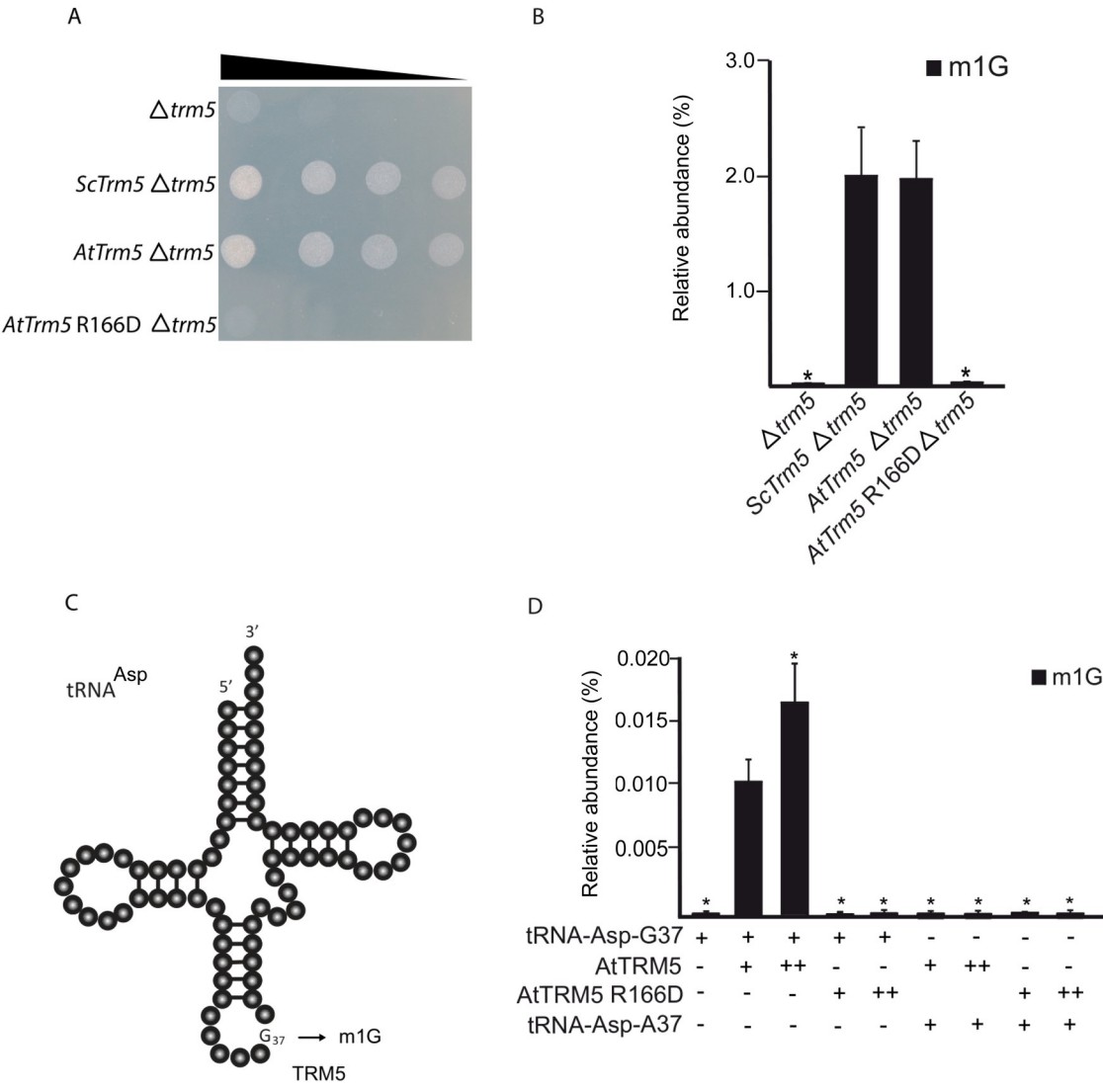

**Fig 4. TRM5 modifies tRNA$_{37}$ guanosine (G) to m$^1$G in yeast.** (**A**) Serial-dilution growth assay. Complementation experiment of TRM5 from yeast (Sc), *Arabidopsis thaliana* (At) or a catalytically inactive mutant AtTrm5 R166D in the yeast *trm5* mutant. (**B**) Relative nucleoside level of modification m$^1$G quantified by HPLC/MS of each complemented yeast strains with either ScTRM5 or AtTRM5. (**C**) Proposed model of TRM5-mediated m$^1$G modification of yeast tRNA-Asp. (**D**) Relative nucleoside level of modification m$^1$G with varying conditions of tRNA-Asp-G37, AtTRM5, AtTRM5 catalytic mutant, and tRNA-Asp-A37. + indicates presence, ++ indicates two-fold increase,—indicates absence.

activity of AtTRM5 was measured by incubating AtTRM5 with the tRNA-Ala-I37 substrate. Previously, in yeast ScTAD1 was demonstrated to deaminate tRNA-Ala-A37 to tRNA-Ala-I37 *in vitro* [26] and *Arabidopsis thaliana tad1* mutants were reported to have reduced tRNA-Ala-I37 [11]. We expressed *AtTAD1* as a GST fusion protein and purified the recombinant GST-AtTAD1 protein. We also generated a catalytically inactive GST-AtTAD1 mutant recombinant protein by using site-directed mutagenesis and expressed and purified the GST-At-TAD1 mutant fusion protein. In a two-step assay, yeast tRNA-Ala-UGC RNA transcripts were generated by *in vitro* transcription using T7 RNA polymerase, the tRNA transcripts were then incubated with the recombinant fusion proteins in the presence of Mg$^{2+}$ and methyl donor S-adenosyl-methionine (AdoMet), and the m$^1$I nucleoside levels measured (Fig 5B). m$^1$I was

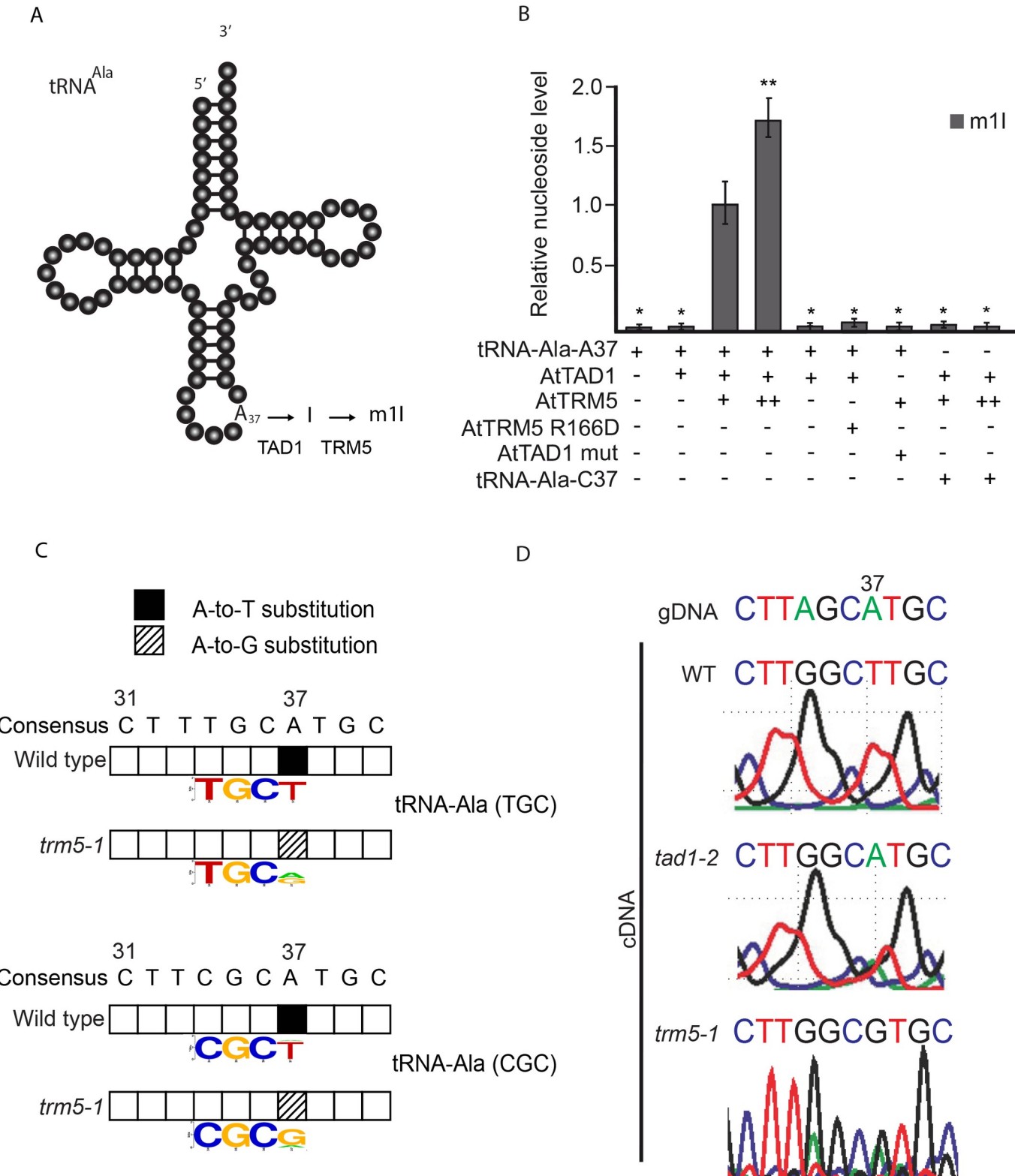

**Fig 5. TRM5-mediated m$^1$I modifications in two-step reaction.** (**A**) Proposed two-step modification model of TRM5-mediated m$^1$G modification of tRNA-Ala. (**B**) Relative nucleoside level of modification m$^1$I with varying conditions of tRNA-Asp-A37, AtTAD1, AtTRM5 mutant, AtTAD1 mutant, and tRNA-Ala-C37. + indicates presence, ++ indicates two-fold increase,—indicates absence. (**C**) Qualitative analysis of tRNA-Ala$^{(TGC)}$ and tRNA-Ala$^{(CGC)}$ modifications. tRNAs were enriched, deep sequenced, aligned using segemehl to tRNA references and the modifications present were inferred from observed

base substitutions between wild type and *trm5-1*. Position 37 in the gDNA, labelled as consensus, is an adenine. Sequence logo shows the proportion of nucleotides and hence inferred modifications at positon 37. Anticodons TGC and CGC are shown in the sequence logos for Ala$^{(TGC)}$ and tRNA-Ala$^{(CGC)}$, respectively. Base substitutions observed from 3 biological replicates are shown. (**D**) Qualitative analysis of tRNA-Ala$^{(AGC)}$ modifications by Sanger sequencing. tRNA-Ala$^{(AGC)}$ was PCR amplified from wild type, *tad1-2* and *trm5-1* and Sanger sequenced. Position 37 in the gDNA is an adenine. In wild-type cDNA, a thymine was detected, in *tad1-2* cDNA an adenine was observed and in *trm5-1* a guanine was detected.

detected only when GST-AtTAD1 and GST-AtTRM5 were provided, and its production occurred in a dosage-dependent manner (Fig 5B). No m$^1$I was detected when the catalytically inactive mutants AtTAD1 E76S or AtTRM5 R166D were provided (Fig 5B). To test the specificity of the methyltransferase activity on tRNA-Ala alanine at position 37, the alanine nucleotide was mutated to a cytosine nucleotide (tRNA-Ala-C37) and the m$^1$I methyltransferase activity was measured. No m$^1$I was detected after incubation with the fusion proteins (Fig 5B). Collectively, these findings interestingly suggest that inosine methylation also occurs at position 37, in addition to guanosine methylation.

To test the AtTRM5 inosine methyltransferase activity *in vivo*, we measured tRNA position modifications by cDNA sequencing from either mutant or wild type plants (Fig 5C and 5D). In the sequencing assay, modification events at position 37 of tRNAs can be directly detected by sequencing of amplified cDNA obtained by reverse transcription and comparison to the DNA reference sequence as inosine is read as guanine (G) and m$^1$I is read as thymine (T) by the reverse transcriptase [10, 35]. As expected, we detected substitutions of A in the reference to T at position 37 for tRNA-Ala-(TGC) and tRNA-Ala-(CGC) by using Illumina sequencing in wild-type plants which is consistent with the presence of m$^1$I (Fig 5C). m$^1$I modifications have previously been described in tRNA-Ala at position 37 in eukaryotes [11, 30, 33]. In the *trm5-1* mutant, no T's were detected at position 37 (Fig 5C) which is consistent with AtTRM5 acting as a tRNA m$^1$I methyltransferase at position 37. Based on our *in vitro* assay results, AtTAD1 first deaminates tRNA-Ala-A37 to tRNA-Ala-I37, and then AtTRM5 methylates tRNA-Ala-I37 to tRNA-Ala-m$^1$I37. We confirmed this pathway in *tad1* and *trm5* mutant plants by Sanger sequencing of tRNA-Ala-(AGC) (Fig 5D). As expected, at position 37, A was substituted to T in the wild type, whereas in *trm5* mutants a G and in *tad1* mutants an A were observed (Fig 5D). These sequencing results are consistent with AtTAD1 first deaminating A37 to I37 as previously reported by Zhou, Karcher (11), and AtTRM5 then methylating I37 to m$^1$I. We also attempted to detect the putative loss of m$^1$G in the tRNA-sequencing data, as it has been reported that m$^1$G is prone to be called as a T in sequencing [35]. However, this was not observed in our datasets. Together, our *in vitro* and *in vivo* data provide support for AtTRM5 possessing tRNA m$^1$I methyltransferase activity.

## AtTRM5 is localized to the nucleus

In yeast, ScTRM5 is localized to both the nucleus and mitochondria [23, 36]. Localisation to mitochondria is thought to be important as yeast strains with only nuclear-localized ScTRM5 exhibited a significantly lower rate of oxygen consumption [23]. In order to determine to which subcellular compartment(s) AtTRM5 is localized in *Nicotiana benthamiana*, we fused TRM5 to the Green Fluorescent Protein (GFP) reporter, transiently infiltrated the construct into leaves and performed laser-scanning confocal microscopy to detect GFP fluorescence. To unambiguously identify the nucleus, we stained the cells with DAPI. When we imaged the cells (n = 100), we observed distinct DAPI fluorescence in a single large circular structure per cell, as expected for the nucleus (Fig 6). Next, we imaged the same cells for GFP fluorescence (Fig 6C) and overlayed the DAPI and GFP fluorescence. The two fluorescence signals showed perfect overlap (Fig 6D). We then searched for nuclear localisation signals (NLS) using the

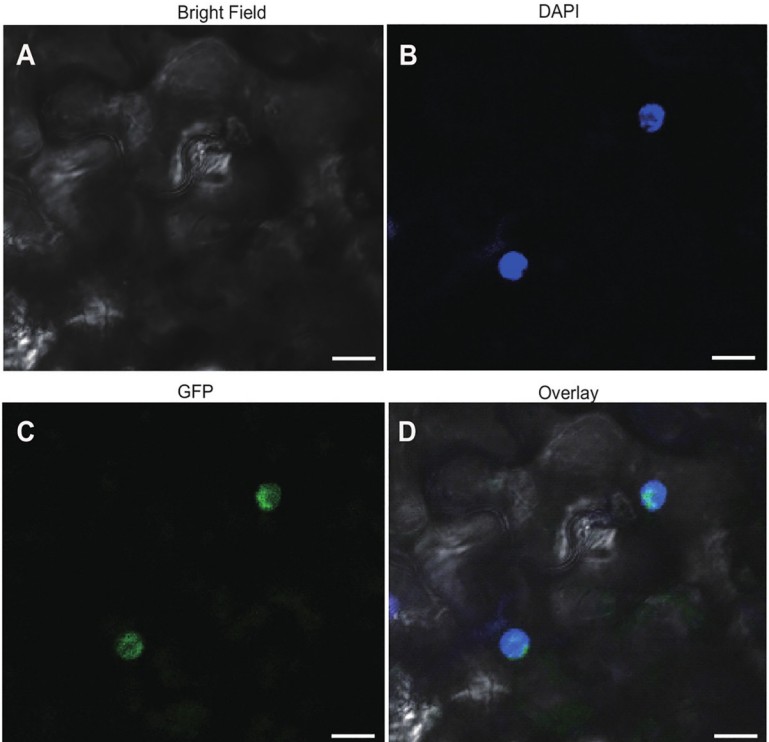

**Fig 6. Subcellular localisation of TRM5 translational reporter protein in *Nicotiana benthamiana* leaves.** TRM5 was fused to Green Fluorescent Protein (GFP) to yield a TRM5-GFP translational fusion recombinant protein. The construct was transiently expressed in *N. benthamiana* leaves and subcellular localisation was determined by confocal laser-scanning microscopy. (**A**) Bright field, (**B**) DAPI stained, (**C**) GFP fluorescence, (**D**) overlay of DAPI and GFP imagines. Scale bars = 20 μm.

LOCALIZER (http://localizer.csiro.au/), LocSigDB (http://genome.unmc.edu/LocSigDB/), and cNLS mapper (http://nls-mapper.iab.keio.ac.jp/cgi-bin/NLS_Mapper_form.cgi) programs [37–39]. While LOCALIZER and LocSigDB did not predict any canonical or bipartite NLS, cNLS mapper predicted with high confidence a 29 amino acid importin α-dependent NLS, QKGCFVYANDLNPDSVRYLKINAKFNKVD, that starts at amino acid 236. Previous finding in yeast suggested that no common canonical or bipartite NLS was detected in ScTrm5 [26], which explains the outcome from LOCALIZER and LocSigDB. Multiple sequence alignment of ScTrm5 and AtTRM5 showed that only a few amino acids were conserved at the region which the importin α-dependent NLS is detected for AtTRM5 (S2 Fig). In summary, we conclude that, unlike in yeast, AtTRM5 in *Arabidopsis* is only localized to the nucleus and may be imported from the cytoplasm into the nucleus by the importin α-dependent pathway.

## Proteins involved in photosynthesis are affected in *trm5* mutant plants

Next, we performed a proteomic analysis to identify proteins that differentially accumulate in *trm5-1* plants when compared to the wild type by using the Tandem Mass Tag (TMT) method (Fig 7). A total of 61571 peptide-spectrum match (PSM) were recorded, corresponding to 29011 peptides and 23055 unique peptides, respectively. 5242 protein groups were identified by blastp searches against the TAIR_pep database. Proteins with fold changes ≥1.20 or ≤0.83 and a significance level of $P \leq 0.05$ were considered to be differentially expressed. In this way, a total of 263 proteins were identified (S3 Table). 102 proteins were upregulated, and 151

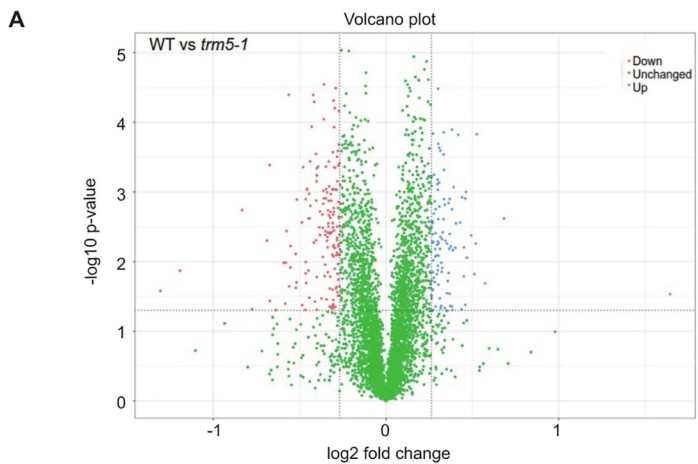

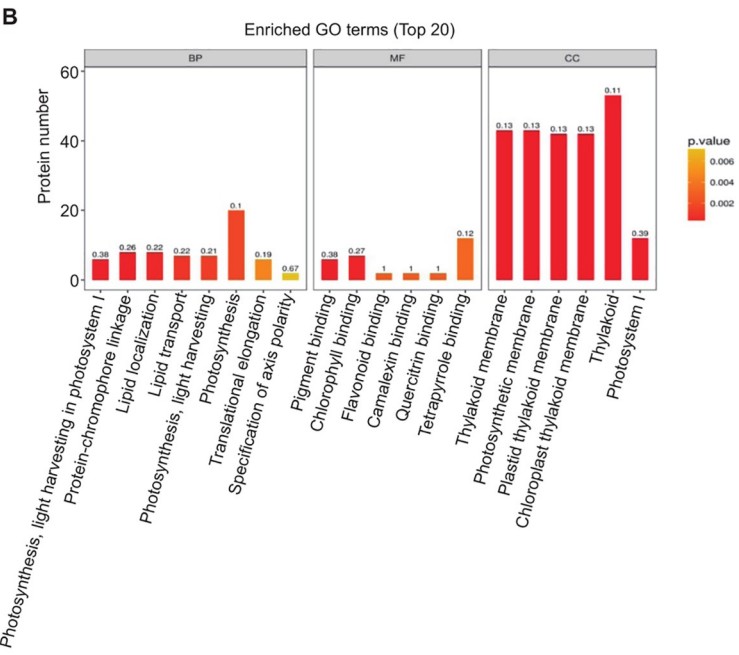

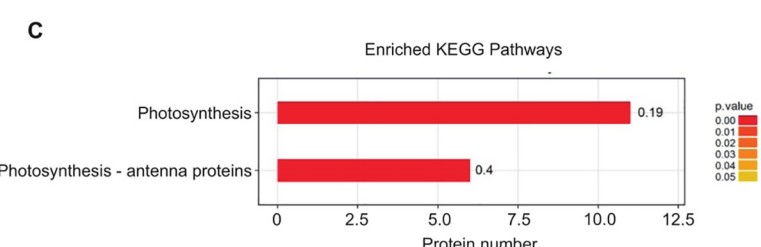

**Fig 7. Proteomic analysis of wild type and *trm5-1*. (A)** Volcano plot of differentially abundant proteins between wild type and *trm5-1*. In *trm5-1*, proteins that were increased in abundance are represented as blue dots, proteins decreased in abundance as red dots, with threshold fold change > 1.2 or < 0.83 (increased or decreased) and *P*-value < 0.05. There were 102 proteins increased and 161 proteins decreased in *trm5-1*. **(B)** GO and **(C)** KEGG terms enrichment analysis. Each differentially abundant protein was first annotated in the GO or KEGG databases, enrichment analysis was performed based on annotated differentially expressed proteins in wild type and *trm5-1*. The top 20 enriched GO terms from Biological Processes (BP), Molecular function (MF), and Cellular Component (CC) are reported.

**Table 1. Comparison between RNA-seq data and proteomics data of selected protein candidates.**

| Gene ID | Protein | log2FC | | Status | |
|---|---|---|---|---|---|
| | | RNA-seq | Proteomics | RNA-seq | Proteomics |
| AT1G03540.1 | Pentatricopeptide repeat (PPR-like) superfamily protein | N/A | 3.12815 | non-differentially expressed | upregulated |
| AT2G45180.1 | Bifunctional inhibitor/lipid-transfer protein/seed storage 2S albumin superfamily protein | 2.012 | 0.56136 | downregulated | downregulated |
| AT2G39480.1 | P-glycoprotein 6 | N/A | 0.43734 | non-differentially expressed | downregulated |
| AT5G50160.1 | ferric reduction oxidase 8 | N/A | 0.40435 | non-differentially expressed | downregulated |

Four protein candidates with the highest fold change (*p-value* < 0.05) reported in the proteomics data were selected for further comparison.

proteins were downregulated in *trm5* (Fig 7A). GO annotation of these differentially accumulating proteins revealed enrichment of the GO terms thylakoid, chloroplast, and photosystem I (Fig 7B). KEGG annotation revealed enrichment of proteins involved in photosynthesis, and photosynthetic proteins, with most of these proteins encoded by genes found in the nuclear and chloroplast genome. (Fig 7C). A further inspection on these GO terms with our proteomics data showed that the photosynthesis-related proteins were downregulated in *trm5* mutant. In the future, the abundance of these photosynthesis-related proteins could be validation by quantitative western blotting analysis. It is worthwhile to state that the abundance of these photosynthesis-related proteins may be directly or indirectly affected by the loss of TRM5 in our experiment. Taken together, the GO and KEGG analysis demonstrated that most differentially accumulating proteins are involved in processes related to photosynthesis.

Defects in tRNA $m^1G$ methylation can be expected to affect mRNA translation, particularly of proteins that have high numbers of affected codons. Therefore, we were interested in identifying genes that showed reduced expression at the protein level in our proteomics analysis of *trm5* plants, but no detectable reduction in mRNA abundance. To identify such mRNAs, we performed RNA-seq on wild type and *trm5* plants. We identified 1186 transcripts that were reduced in abundance in *trm5* and 580 transcripts that were increased in abundance by at least 2-fold and hierarchically clustered these transcripts (S2 Table and S5 Fig). Comparison of the RNA-seq and proteomics datasets identified 133 proteins with reduced abundance, but with no detectable reduction in mRNA abundance (S5 Fig). We further inspected the data by selecting four candidate proteins with the highest fold change reported in the proteomics data (Table 1). From the selected candidate proteins, we discovered that three of the differentially expressed proteins reported in the proteomics data were not differentially expressed in the RNA-seq data, indicating that there is no change in transcripts levels leading to the fluctuation of their corresponding proteins. Only one protein candidate (AT2G45180.1) showed decreased fold change of mRNA, which correlates with the decrease of its corresponding protein reported in the proteomics data.

Finally, we performed a codon usage frequency analysis for this class of genes and detected no codon bias towards triplets read by both $m^1I$ and $m^1G$ modified tRNAs (S5 Fig).

## Modifications at position 37 affects tRNA-halve abundance

Since our previous results showed no bias towards codon usage in *Attrm5* mutants when compared to wild type, we then asked if depleting $m^1G37$ and $m^1I37$ affects tRNA abundance. To test this, we performed a northern blot analysis on the AtTRM5 substrates tRNA-Ala-(AGC) and tRNA-Asp-(GTC), respectively (Fig 8A). We did not observe any substantial change in the

A

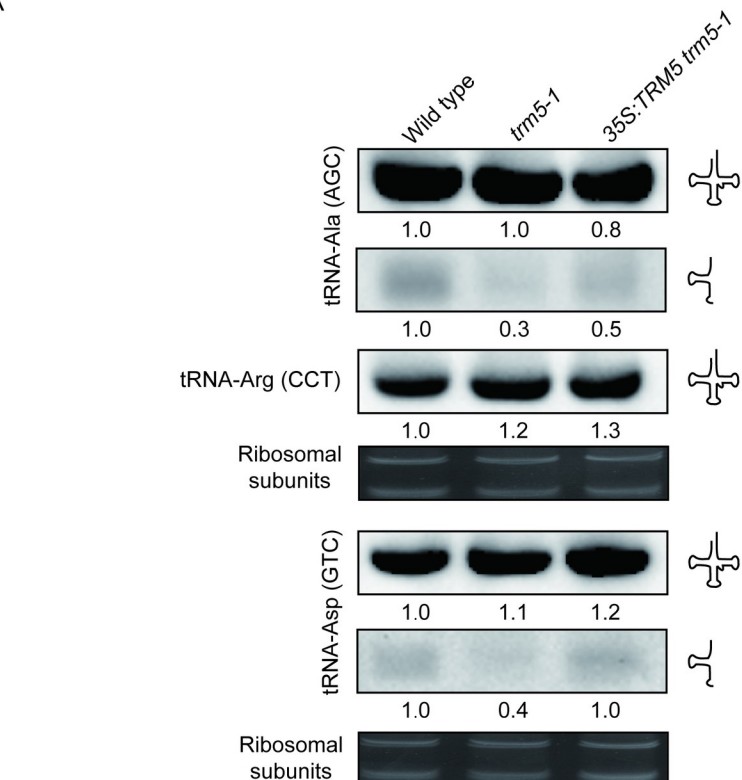

B

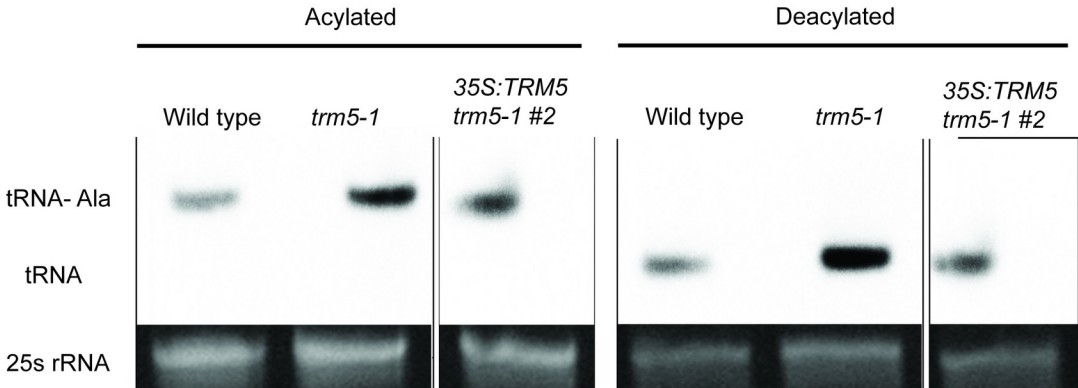

**Fig 8. Loss of function of AtTRM5 affects tRNA halve abundance.** (**A**) Northern blot analysis of RNA from wild type, *trm5-1* and *35S:TRM5 trm5-1*. tRNA-Ala (AGC) was inspected for changes in m$^1$G37 and tRNA-Asp (GTC) and m$^1$I37. tRNA-Arg (CCT) and rRNA were used as loading controls. The blots were probed with either tRNA-Arg (CCT), tRNA-Asp (GTC), or tRNA-Ala (AGC). (**B**) Accumulation and aminoacylation of tRNA-Ala (AGC) in wild-type plants, *trm5-1* mutant plants and 35S:*TRM5 trm5-1*. The aminoacylated tRNA migrates slower than its corresponding deacylated species. To visualize the difference in electrophoretic mobility, aliquots of the same samples were deacylated *in vitro*. 10 μg of RNA was separated by electrophoresis, blotted and hybridized to a tRNA-Ala (AGC)-specific probe. The 25S rRNA band of the ethidium bromide-stained gel prior to blotting is shown as a loading control.

full-length tRNA abundances for both tRNA-Ala-(AGC) and tRNA-Asp-(GTC). However, we unexpectedly observed significant decreases in the 5' half abundance of both tRNA-Ala-(AGC) and tRNA-Asp-(GTC) (Fig 8A), suggesting that the loss of m$^1$G37 and m$^1$I37 affects tRNA halve steady state abundance.

The anticodon region of tRNAs are known to impact tRNA aminoacylation as it interacts with tRNA aminoacyl synthetase [40] and that tRNA position 37 at the 3' anticodon region has a significant role in facilitating tRNA structure fidelity and anticodon interactions [20]. Therefore, we investigated if the tRNA position 37 which is proximal to the 3' anticodon could affect tRNA aminoacylation for tRNA-Ala-(AGC). We performed a tRNA aminoacylation experiment to determine if the loss of m$^1$G37 modification at the anticodon region of tRNA-Ala-(AGC) affects tRNA charging (Fig 8B) and our results show that the loss of function of *AtTRM5* does not affect acylation of tRNA-Ala (Fig 8B). One explanation for this observation can be attributed to the nature of alanyl-tRNA synthetases that recognizes G3:U70 of tRNA acceptor stem for tRNA charging instead of the anticodon region [41]. This showed that the loss of m$^1$G37 modification does not inhibit aminoacylation.

## Discussion

The discovery of m$^1$G and m$^1$I at position 37 in tRNAs of a wide range of eukaryotic and prokaryotic organisms underscores its importance as a key regulator of tRNA function and, presumably, translation [10, 21, 23]. Previous studies in bacteria have shown that m$^1$G37 is required for translational fidelity [21, 42, 43] and that mutations in the enzymes catalysing m$^1$G37 severely impact growth or cause lethal phenotypes [10, 23, 31]. The presence of m$^1$G37 modification on tRNA-Asp prevents erroneous amino-acylation by arginyl-tRNA synthetase as both tRNA-Asp and tRNA-Arg have highly similar structures [44].

In eukaryotes, m$^1$G37 modification requires the methyltransferase TRM5. Here we report that in plants, *trm5* mutants have a 60% reduction in m$^1$G and m$^1$I levels, display severely reduced growth and delayed transition to flowering. This is somewhat similar to human patients that are heterozygous for one mutant and one functional allele of *Trm5*. They show childhood failure to thrive and exercise intolerance symptoms [10].

With the 60% reduction of m$^1$G and 75% reduction of m$^1$I in *trm5* plants, it is reasonable to anticipate a significant impact on protein translation due to the reduced stacking effect of m$^1$G37 on base-pairing between position 36 and the first nucleoside in translated mRNA codons. We observed a reduction in abundance of proteins involved in ribosome biogenesis and photosynthesis, and this is likely to account for the observed reduced growth of the mutant plants. A similar study by Jin *et al.* (2019) showed higher levels of polysomes in *attrm5a* mutants compared to wild type and decreased levels of ribosomal subunit, which is anticipated to reduce translation rate [45]. In the future, it would be interesting to test if translational errors, such as ribosome stalling or frame shifting, occur in *trm5* plants which would be similar to what has been previously reported in bacteria with reduced tRNA m$^1$G [26].

We localized *Arabidopsis* TRM5 to the nucleus which is in contrast to the cytoplasmic and mitochondrial localisation in other eukaryotes: *Trypanosoma brucei*, *Homo sapiens* (HeLa cells) and *Saccharomyces cerevisiae* [10, 23, 31], although in one study *S. cerevisiae* Trm5 has also been localized exclusively to the nucleus [36]. Interestingly, in yeast, Trm5 acts in the nucleus to form m$^1$G on retrograde imported tRNA$^{Phe}$ after initial export from the nucleus and subsequent splicing at the mitochondrial outer membrane. As retrograde tRNA import is conserved from yeast to vertebrates [46, 47], it is tempting to speculate that TRM5-mediated m$^1$G formation occurs in retrograde imported tRNAs in *Arabidopsis*.

Direct comparison of RNA-seq and proteomics data has been challenging as variation in protein abundance in proteomics datasets can be confounded by multiple factors. Several factors can cause fluctuation in protein levels with no change of mRNA abundance: mRNA transcript abundance, translation rate, and translation resources availability (as reviewed in Liu, Beyer (48)). This may explain the results of the initial comparison between our RNA-seq and proteomics data, and the seeming discrepancies in the derived sets of differentially expressed genes (S2 and S3 Tables). In our RNA-seq data, microtubule-related genes appeared to be significantly downregulated (AT1G21810, AT1G52 410, AT2G28620, AT2G44190, AT3G23670, AT3G60840, AT3G63670, AT5G67270, AT4G14150, AT5G27000). However, we could not detect any differential expression of these proteins in our proteomics dataset. This is not unexpected, as it has been reported that protein and transcript abundances only rarely correlate in data analysis [48]. However, it has been shown that transcript abundance can still be used to infer protein abundance [49]. Among the selected four candidate proteins (Table 1), we could observe only one event of protein downregulation due to decreased transcript level: the lipid transfer protein (AT2G45180) which is involved in lipid transport in chloroplasts. Lipid transport is crucial in the formation of photosynthetic membranes in plants [50]. As several lipids are important functional components of thylakoidal protein complexes involved in photosynthesis, the impacted lipid transport and lipid synthesis revealed by the GO term analysis may contribute to the observed dysregulation of photosynthesis-related genes. This finding is consistent with Jin *et al.* (2019) [45]. The remaining three candidate genes displayed no changes in transcript levels, but one candidate gene showed upregulation and the other two downregulation at the protein level. The exact function of pentatricopeptide repeat (PPR-like) superfamily protein (AT1G03540.1) is unknown, but the protein was suggested to play a significant role in post-transcriptional modification of tRNAs [51]. The P-glycoprotein 6 (AT2G39480) belongs to the large P-glycoprotein family and is known to play a role in mediating auxin transport, and disturbed auxin transport is known to affect plant growth [52]. Our proteomics data suggested that the auxin transport is affected (AT5G35735, AT3G07390, AT4G12980, AT5G35735) in the *trm5* mutant. This can be inferred to the reduced auxin levels in *attrm5a* as reported by Jin *et al.* (2019) [45]. Ferric reduction oxidase 8 (AT5G50160/AtFRO8) has been shown to participate in iron reduction and was implicated in leaf vein transport [53]. We proposed that disturbed photosynthesis in *trm5* mutant plants is a secondary effect of dysregulated transport processes.

Our findings also show that modified bases at tRNA-Ala at position 37 does not affect tRNA aminoacylation but does affect tRNA-Ala-derived 5' half steady state abundance. Both tRNA aminoacylation and tRNA-derived halves were shown to effect translational fidelity [40, 54]. In yeast, the introduction of tRNA-derived halves, both 5' and 3' fragments, affects tRNA aminoacylation, suggesting that tRNA-derived fragments (tRFs) has a regulatory role in translation [54]. It was also demonstrated that these tRFs regulate tRNA-aminoacylation via interacting with ribosome. The decreased amount of tRNA-Ala 5' halves may explain the polysome profile changes reported by Jin *et al.* (2019) [45].

The role of post-transcriptional RNA modifications in tRNA and mRNA metabolism and their impact on plant growth and development in plants are only beginning to be elucidated. Here, we have described the AtTRM5-mediated $m^1G$ and $m^1I$ methylation in tRNAs and identified crucial links between this modification, photosynthesis, plant growth, and protein translation. It appears likely that the many other tRNA modifications in plant tRNAs also play important roles in translation and/or translational regulation which remain to be discovered.

## Materials and methods

### Plant material and root growth experiments

*Arabidopsis thaliana* (Columbia accession) wild type and mutant plants were grown in Phoenix Biosystems growth under metal halide lights as previously described [55]. For plate experiments, seeds were first surface sterilized, plated on ½ MS medium supplemented with 1% sucrose and sealed as previously described [1, 56]. All plants were grown under either long-day photoperiod conditions of 16 h light and 8 h darkness or short-day photoperiods of 10 h light and 14 h darkness.

Characterization of the mutant alleles, *trm5-1* (SALK_022617) and *trm5-2* (SALK_032376) are as described previously [57]. The *tad1-2* mutant was used as described previously [11]. Nucleotide sequence data for the following genes are available from The *Arabidopsis* Information Resource (TAIR) database under the following accession numbers: TRM5 (At3g56120), At4g27340, At4g04670, and TAD1 (At1g01760).

Analysis of root phenotypes was carried out on 11-day-old seedlings grown on ½ MS agar plates. A flatbed scanner (Epson) was used to non-destructively acquire images of seedling roots grown on the agar surface. Once captured, the images were analysed by software package RootNav [58, 59].

### Plasmid construction and generation of transgenic plants

For the 35SCaMV:TRM5 construct, the full-length genomic region of At3g56120 including the 5'UTR and 3'UTR was amplified from Col-0 genomic DNA template with primers provided in (S1 Table) and cloned into Gateway entry vector pENTRTM/SD/D-TOPO (Invitrogen). The insert was sequenced and then cloned into the binary destination vector pGWB5 by an LR recombination reaction, using the Gateway cloning system following the manufacturers protocol (Invitrogen), resulting in the 35S:TRM5 construct. For the TRM5$_{Pro}$::TRM5 construct, the full-length genomic region of At3g56120 that included the promoter, 5'UTR and 3'UTR was amplified from Col-0 genomic DNA template with primers provided in (S1 Table) and cloned into Gateway entry vector PCR8 TOPO-TA (Invitrogen). The insert was sequenced and then cloned into the a modified (35S promoter removed) destination pMDC32 vector, using the Gateway cloning system [60] following the manufacturers protocol (Invitrogen), resulting in TRM5$_{Pro}$::TRM5. The 35S:TRM5 construct was transformed into *A*. *thaliana* wild type Col-0 plants or *trm5-1* mutant plants by *Agrobacterium*-mediated floral dip method respectively [61]. The TRM5$_{Pro}$::TRM5 construct was transformed into *trm5-2* mutant plants by *Agrobacterium*-mediated floral dip method. Transgenic plants were selected on ½ MS media supplemented with 50 µg ml$^{-1}$ kanamycin. TRM5 transcript abundance was assessed in at least five independent T$_1$ plants using qRT-PCR and two lines showing the highest TRM5 transcript levels were carried through to homozygous T$_3$ generation for phenotypic analysis.

### Sub-cellular localization of TRM5

For analysis of subcellular localization of TRM5, a 35S:TRM5:GFP construct was produced by PCR amplification of the TRM5 coding sequence from *Arabidopsis* seedling cDNA by using the primers described in the supplementary data (S1 Table). The TRM5 cDNA was recombined cloned into pCR8, sequenced and then recombined in frame into pMDC83 to produce 35S:TRM5:GFP. The 35S:TRM5:GFP construct was introduced into *A*. *tumefaciens* strain GV3101 and transiently expressed in 5-week-old *Nicotiana benthamiana* leaves. Fluorescence was analysed using a confocal laser-scanning microscope (Zeiss microscope, LSM700) and excited with 488-nm line of an argon ion laser. GFP fluorescence was detected via a 505- to

530-nm band-pass filter. The cut leaves were immersed in 10 μM 4′, 6-diamidino-2-phenylin-dole (DAPI) at room temperature for 45 min, and then washed with PBS for 3 times (5 min each). The blue fluorescence of DAPI was imaged using 404-nm line for excitation and a 435- to 485-nm band pass filter for emission.

## Shoot apical meristem sections

14, 18, or 22-day-old seedlings were fixed for 1 day in FAA containing 50% ethanol, 5% acetic acid and 3.7% formaldehyde. The samples were then dehydrated through an ethanol series of five one-hour steps (50, 60, 70, 85, 95% ethanol) ending in absolute ethanol (100%). The ethanol was gradually replaced with Histoclear containing safranine to stain the tissue, as following: incubated in a Histoclear series (75:25, 50:50, 25:75 Ethanol: Histoclear) for 30 min and followed by twice one-hour incubation in 100% Histoclear. The paraffin was polymerised by baking overnight at 60°C, and the samples were embedded in paraffin. Sections were cut, attached to slides and dried on a slide warmer overnight at 42°C to allow complete fixation. The shoot apical meristem was observed using light microscopy.

## Quantitative real-time PCR (qPCR)

For the transcription profiling of flowering-related genes and circadian clock-related genes, 17-day-old seedlings were sampled from Zeitgeber time (ZT) 1 and collected every 3h during the day and night cycles, respectively. Total RNA was extracted the leaf samples using Trizol reagent (Invitrogen). The relative expression levels of *AtTRM5* were determined using quantitative real-time PCR (qPCR) with gene-specific primers (S1 Table). The qPCR was performed using the StepOnePlus real-time PCR system (Applied Biosystems) using Absolute SYBR Green ROX mix (Applied Biosystems) for quantification. Three biological replicates were carried out for each sample set. The relative expression was corrected using a reference gene *EF1alpha* (At5g60390) and calculated using the $2^{-\Delta\Delta Cq}$ method as described previously [11].

## mRNA-sequencing

Total RNA, 1 ug, was extracted from 20-day-old *Arabidopsis* leaf samples using Trizol reagent (Invitrogen) and purified using the RNAeasy Mini RNA kit (Qiagen). One hundred nanograms of RNA were used for RNA-seq library construction according the manufacturer's recommendations (Illumina). First-strand cDNA was synthesized using SuperScript II Reverse Transcriptase (Invitrogen). After second strand cDNA synthesis and adaptor ligation, cDNA fragments were enriched, purified and then sequenced on the Illumina Hiseq X Ten. Three biological replicates were used for RNA-seq experiments.

## tRNA purification and tRNA-sequencing

Total RNA was isolated from wild type and *trm5* 10-day-old *Arabidopsis* seedlings using the Spectrum Plant total RNA kit (SIGMA-ALDRICH) and contaminating DNA removed using DNase I (SIGMA-ALDRICH). To enrich for tRNAs, 10μg of total RNA was separated on a 10% polyacrylamide gel, the region containing 65–85 nts was removed and RNA was purified as previously described [56]. Purified tRNAs were used for library construction using NEB Ultradirectional RNA library kit. Given the short sequences of tRNAs, the fragmentation step of the library preparation was omitted, and samples were quickly processed for first-strand cDNA synthesis after the addition of the fragmentation buffer. The remaining steps of library construction were performed as per the manufacturer's instructions. Illumina sequencing was

performed on a MiSeq platform at The Australian Cancer Research Foundation (ACRF) Cancer Genomics Facility, Adelaide.

## Yeast complementation

AtTrm5 (At3g56120) and ScTrm5 (YHR070W) was PCR amplified from cDNA and cloned into pYE19 using a Gibson assembly reaction (NEB). Mutant AtTrm5 (R166D) was generated by synthesising gene blocks (IDT) with nucleotides that mutated the translated proteins at R166 and the gene block was cloned into pYE19 using a Gibson assembly reaction (NEB). Yeast △trm5 (*Mat a*, *hisD1*, *leu2D1*, *met15D0*, *trm5:KanMX)* was previously described [62]. Recombinant plasmids were transformed into △*trm5* mutant strain and the resulting strain was analysed for growth phenotypes and m$^1$G nucleoside levels.

## AtTrm5, AtTAD1, ScTrm5 protein expression and purification and tRNA methylation

Full length AtTrm5, mutant AtTrm5 (R166D) and AtTAD1 (At1g01760) cDNAs were cloned into pGEX resulting in GST-AtTrm5, GST-AtTRM5-mutant, GST-TAD1, respectively. Mutant AtTAD1 (E76S) was generated by synthesising a gene block (IDT) with mutated nucleotides and the gene block was cloned into pGEX using a Gibson assembly reaction (NEB) resulting in GST-TAD1 mutant. IPTG (0.5 μM) was used to induce expression of the proteins and the recombinant proteins were purified on a GST resin column (ThermoFisher Scientific). tRNA-Asp-GUC or tRNA-Ala-AGC were transcribed in vitro with T7 RNA polymerase (Promega). Methylation reactions were performed in 100 mM Tris-HCl, 5 mM MgCl$_2$, 100 mM KCl, 2 mM DTT, 50 mM EDTA, 0.03 mg/mL BSA and 25 μM AdoMet. Substrate tRNA was provided in a final concentration of 1–5 μM, AtTrm5 or AtTrm5 mutant proteins from 6.0 to 12 μM and AtTAD1 or AtTAD1 mutant proteins at 5.0 μM.

## mRNA-seq and tRNA-seq bioinformatics analysis

Global Mapping: Reads were first adapter trimmed using Trim galore v0.4.2 (http://www.bioinformatics.babraham.ac.uk/projects/trim_galore/) with default parameter. The quality of the trimmed reads was checked using Fastqc (https://www.bioinformatics.babraham.ac.uk/projects/fastqc/) and ngsReports (https://github.com/UofABioinformaticsHub/ngsReports). Reads were then globally mapped to *Arabidopsis* reference genomes (TAIR10) [63] using STAR v2.5.3 [64]. Mapped reads were counted using featureCounts [65]. The raw read counts were normalized using sample size factor for sequence depth and differential expression analysis between wild type and *trm5-1* mutant samples was performed using DESeq2 v1.18.1 [66]. The tRNA-enriched reads were mapped to tDNA reference acquired from GtRNAdb (generated from tRNA-Scan SE v2.0) of *Arabidopsis thaliana* derived from TAIR10 reference genome [63, 67–69] using segemehl v0.2.0 [68]. The mapped reads were then processed for variant calling using GATK v3.7 [70]. Haplotype Caller in GATK v3.7 was implied to call for variants, with the variant filtered using hard filtering as recommended.

Proportion estimation: SNPs were identified in the wild type and *trm5-1* samples were then compared. SNPs at position 37 were extracted and analysed using vcftools [71]. Changes in base pair modifications were indicated by base substitution due to the property of next generation sequencing as mentioned in [10, 35]. The ratio of the expected A-to-T conversion in wild type samples and both the ratio of A-to-T (indicating no change in comparison to wild type) and A-to-G in *trm5-1* were analyzed as an indication of m$^1$I depletion.

## Sanger sequence analysis of tRNA editing

tRNA purification and tRNA editing analysis were performed as described previously [11]. Cytosolic tRNA-Ala (AGC) were amplified by reverse transcriptase (RT)-PCR with specific primers (tRNA-Ala-f and tRNA-Ala-r, S1 Table), and purified PCR products were directly Sanger sequenced.

## TMT-based proteome determination and data analysis

Total protein was extracted from 20-day-old *Arabidopsis* leaf samples and purified according to a method described by [72]. Protein digestion was performed according to FASP procedure described previously [73], and 100 ụg peptide mixture of each sample was labelled using TMT reagent according to the manufacturer's instructions (Thermo Fisher Scientific). LC-MS/MS analysis was performed on a Q Exactive mass spectrometer (Thermo Fisher Scientific) that was coupled to Easy nLC (Thermo Fisher Scientific). MS data was acquired using a data-dependent top20 method dynamically choosing the most abundant precursor ions from the survey scan (200–1800 m/z) for HCD fragmentation (Shanghai Applied Protein Technology Co., Ltd). Determination of the target value is based on predictive Automatic Gain Control (pAGC). SEQUEST HT search engine configured with Proteome Discoverer 1.4 workflow (Thermo Fisher Scientific) was used for mass spectrometer data analyses. The latest *Arabidopsis* protein databases (2018) was downloaded from http://www.arabidopsis.org/download/index-auto.jsp?dir=/download_files/Proteins and configured with SEQUEST HT for searching the datasets. The following screening criteria was deemed as differentially expression: fold-changes >1.2 or <0.83 (up- or down-regulation) and the Benjamini-Hochberg corrected *p*-value, $P < 0.05$ were considered significant.

## tRNA nucleoside analysis

tRNAs were purified as previously described [56]. Twenty-five μg of tRNAs were digested with P1 nuclease (Sigma- Aldrich) and 1.75 units of calf intestine alkaline phosphatase in 20 mM Hepes–KOH (pH 7.0) at 37˚C for 3 h. The mixture was diluted to a concentration of 15 ng/uL and samples were injected into an API 4000 Q-Trap (ThermoFisher Scientific) mass spectrometer coupled with LC-20A HPLC system (Shimadzu). All RNA samples analyzed were from three biological replicates. A diode array UV detector monitored the LC signals from the nucleosides, whereas in counts were recorded in positive mode (190–400 nm). The abundance of each modified nucleoside was represented by its unique ion peak area and normalized to the sum of the four canonical nucleosides (A, C, G and U nucleosides). Since $m^1G$, $m^2G$ and $m^7G$ have the same Q1 and Q3 mass, they were discriminated by retention time, in the order of $m^7G$, $m^1G$ and $m^2G$, respectively. The identity of $m^7G$ peak was confirmed with external standard (TriLink Biotechnologies) and the identity of $m^1G$ versus $m^2G$ was indirectly confirmed by results from yeast *trm5* mutant

## Northern blot for tRNA detection

Total RNA was isolated from the wild type, *trm5-1* and *35S:TRM5 trm5-1 Arabidopsis* seedlings by using the TRIZOL Reagent (ThermoFisher Scientific). 20μg of total RNA was on 15% urea-polyacrylamide gel and RNA was transferred to Amersham Hybond-N⁺membrane. After cross-linking, prehybridization was performed for 3 hours at 68˚C in 20μl of DIG easy Hyb Granules (Roche). 60 pmol of 3-end Digoxin labelled oligonucleotides was added to hybridization buffer (S1 Table) and hybridize for overnight at 40˚C with gentle agitation. The membrane was washed at low-stringency (2X SSC, 0.1% SDS) twice and high-stringency (0.1X SSC,

0.1% SDS) twice. The probe-target hybrid was localised with the anti-DIG-alkaline phospha-tase antibody (Roche), and the membrane was put into washing and blocking and detection buffer to visualize using the CDP-Star, ready to use (Roche).

### tRNA aminoacylation assay

Total RNA of wild type, *trm5-1* and *35S:TRM5 trm5-1 Arabidopsis* seedlings were isolated and analysed using the tRNA aminoacylation protocol as described in Zhou, Karcher (11), using a specific probe for tRNA-Ala (AGC) (S1 Table).

## Supporting information

**S1 Fig. Extended phylogenetic analysis of TRM5 orthologues.** (**A**) Circos plot of sequence conservation of TRM5 orthologues in yeast (Sc), tomato (Sl), grape (Vv), *Arabidopsis* (At), maize (Zm), rice (Os), *Marchantia* (Mp), *Physcomitrella* (Pp), *Chlamydomonas* (Cr),*Ostreococ-cus* (Ot), humans (HsTrm5), *Drosophila melanogaster* (DmTrm5), *Pyrococcus horikoshii* (PhTYW2), and *Methanococcus jannaschii* (MjTYW2). The ribbons were coloured based on sequence identity, with blue < = 25%, green 25–50%, orange 51–75% and red for 76–99%. (**B**) Unrooted phylogenetic tree of the same TRM5 orthologues used to for sequence conservation analysis.
(TIF)

**S2 Fig. Bioinformatics characterization of TRM5 and related proteins.** (**A**) Multiple sequence alignment of TRM5 proteins from *Arabidopsis thaliana* (At), yeast (ScTrm5), humans (HsTrm5), *Drosophila melanogaster* (DmTrm5), *Pyrococcus horikoshii* (PhTYW2), and *Methanococcus jannaschii* (MjTYW2). Black shaded boxes are identical across all species. Light shaded boxes are similar and nearly conserved residues. Asterisk indicates catalytic important amino acids. The predicted 29 aa importin α-dependent NLS is boxed in red. (**B**) Multiple sequence alignment of yeast Trm5p, *Arabidopsis* TRM5 (At3g56120) and the two closest related proteins from *Arabidopsis*. Black shaded boxes are conserved in at least 2 sequences. Met 10+ like domain and S-adenosyl Methyltransferase Motif (SAM) are detected in *Arabidopsis* TRM5 using NCBI Conserved Domains Search (https://www.ncbi.nlm.nih.gov/Structure/cdd/wrpsb.cgi). The position of importin α-dependent NLS is also indicated.
(TIF)

**S3 Fig. Characterisation of growth and development in wild type, trm5 mutant, comple-mented and overexpression lines.** (**A**) Seeds (n = 100) of wild type and *trm5-1* were sown on ½ MS plates, stratified at 4 °C and then grown at 21 °C under long day conditions for 32 hours. Germination was measured at 8, 16, 24 and 32 hours after shifting to 21 °C. (**B**) Sections of the shoot apical meristems of wild type and *trm5-1* plants grown under long days for 14, 18 and 22 days. (**C**) The average fresh plant weight of long day grown plants. (**D**) The average rosette leaf number at flowering; (**E**) The average days to flowering under long days. (**F**) The rosette leaf number under short days. Data presented are means. Error bars are ± SE (n = 16). NF = did not flower. An asterisk indicates a statistical difference ($P<0.05$) as determined by Student's t-test.
(TIF)

**S4 Fig. Root phenotypic analysis of seedlings.** Seedlings of wild type, *trm5*, complemented lines (35S:TRM5 *trm5-1*), TRM5 overexpression lines (35S:TRM5) were vertically grown on ½ MS medium for 10 days and then measured. (**A**) Total root length, (**B**) Primary root, (PR) length, (**C**) average lateral root (LR) length and (**D**) LR number were measured 10 days after

germination. Data presented are means. Error bars are ± SE (n = 10 plants).
(TIF)

**S5 Fig. RNA-seq and proteomic analysis of wild type and trm5-1.** (**A**) RNA was purified from 10-day-old seedlings of wild type (wt) and *trm5-1* (n = 3). RNA-seq analysis was performed and differentially abundant transcripts were hierarchically clustered. (**B**) An upset plot showing positive overlapping mRNAs and proteins identified by RNA-seq and proteomics analysis. (**C**) Codon bias analysis of codons in the up and down regulated proteins identified by proteomics analysis.
(TIF)

**S1 Table. Oligonucleotide primers used in this study.**
(DOCX)

**S2 Table. Differentially expressed genes identified by RNA-seq analysis of wild type and *trm5-1*.**
(XLSX)

**S3 Table. Differentially abundant proteins identified by LC-MS/MS analysis of wild type and *trm5-1*.**
(XLSX)

## Acknowledgments

We thank the staff at The Australian Cancer Research Foundation (ACRF) Cancer Genomics Facility, Adelaide for their technical expertise in sequencing. We would also like to thank the staff members of the University of Adelaide Bioinformatics Hub for bioinformatics consultation.

## Author Contributions

**Conceptualization:** Wenbin Zhou, Iain Searle.

**Formal analysis:** Qianqian Guo, Pei Qin Ng, Shanshan Shi, Diwen Fan, Jun Li, Jing Zhao, Hua Wang, Rakesh David, Parul Mittal, Trung Do, Ralph Bock, Wenbin Zhou, Iain Searle.

**Funding acquisition:** Ming Zhao, Iain Searle.

**Investigation:** Qianqian Guo, Iain Searle.

**Methodology:** Wenbin Zhou, Iain Searle.

**Project administration:** Iain Searle.

**Supervision:** Iain Searle.

**Writing – original draft:** Pei Qin Ng, Iain Searle.

**Writing – review & editing:** Qianqian Guo, Pei Qin Ng, Jing Zhao, Hua Wang, Rakesh David, Trung Do, Ralph Bock, Ming Zhao, Wenbin Zhou, Iain Searle.

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
