## [Decision Letter · Decision Letter 0]

12 Aug 2019

PONE-D-19-17540

Arabidopsis TRM5 Encodes a Nuclear-localised Bifunctional tRNA Guanine and Inosine-N1-methyltransferase that is Important for Growth

PLOS ONE

Dear Dr Searle,

Thank you for submitting your manuscript to PLOS ONE. After careful consideration, we feel that it has merit but does not fully meet PLOS ONE’s publication criteria as it currently stands. Therefore, we invite you to submit a revised version of the manuscript that addresses the points raised during the review process.

Your manuscript has been evaluated by two experts in the field. While their comments are in general positive, they have also raised a number of points that need to be fixed and/or clarified (e.g. by giving additional details of your experimental and data analysis protocols) before your manuscript can be accepted for publication in PLOS ONE. I have also taken the time to read your manuscript carefully, and my understanding is that most of those points are relatively easy to address, either in the main body of your manuscript or in the rebuttal letter, as appropriate. 

I understand that you want to keep the focus of your work on only one of the three paralogous genes, and I am not asking you to perform additional experiments regarding this. However, please make sure to temper your conclusions, as suggested by reviewer #2 in point #1 of his/her evaluation report. This reviewer has also pointed out that some of your data have not been made available (i.e. Figure S6); please make sure that all figures and other data supporting your conclusions are available in the revised version, in line with the journal's policy on data availability.

We would appreciate receiving your revised manuscript by Sep 26 2019 11:59PM. To enhance the reproducibility of your results, we recommend that if applicable you deposit your laboratory protocols in protocols.io, where a protocol can be assigned its own identifier (DOI) such that it can be cited independently in the future. For instructions see: http://journals.plos.org/plosone/s/submission-guidelines#loc-laboratory-protocols

We look forward to receiving your revised manuscript.

Kind regards,

Hector Candela, Ph.D.

Academic Editor

PLOS ONE

Journal Requirements:

1. We note that you have stated that you will provide repository information for your data at acceptance. Should your manuscript be accepted for publication, we will hold it until you provide the relevant accession numbers or DOIs necessary to access your data. If you wish to make changes to your Data Availability statement, please describe these changes in your cover letter and we will update your Data Availability statement to reflect the information you provide.

2. Please amend your list of authors on the manuscript to ensure that each author is linked to an affiliation. Authors’ affiliations should reflect the institution where the work was done (if authors moved subsequently, you can also list the new affiliation stating “current affiliation:….” as necessary).

Reviewers' comments:

Reviewer's Responses to Questions

**Comments to the Author**

1. Is the manuscript technically sound, and do the data support the conclusions?

Reviewer #1: Yes

Reviewer #2: Partly

2. Has the statistical analysis been performed appropriately and rigorously? 

Reviewer #1: Yes

Reviewer #2: I Don't Know

3. Have the authors made all data underlying the findings in their manuscript fully available?

Reviewer #1: Yes

Reviewer #2: No

4. Is the manuscript presented in an intelligible fashion and written in standard English?

Reviewer #1: Yes

Reviewer #2: No

5. Review Comments to the Author

Reviewer #1: This manuscript (PONE-D-19-17540) reports the characterization of Arabidopsis TRM5 and the growth phenotype of Arabidopsis trm5 gene disruptant strain. TRM5 is a tRNA methyltransferase, which methylates G37 and I37 in tRNA, and produces m1G37 and m1I37, respectively. As described in this manuscript, there are little knowledges about tRNA modification enzymes in plant. Although Jin et al. reported similar characterization of Arabidopsis TRM5 during the course of study by the authors, I believe that this manuscript contains several scientific merits to be published in the journal “PLOS ONE”. Before the publication, however, I would like to suggest some minor revisions.

Minor points:

(1) Page 6 lines 139-143.

As far as I know, m1I is found only at position 37 in tRNAs from eukaryotes.

However, m1I in tRNAs from trm5 mutants is not lost completely. I feel that some explanations for this observation may be necessary.

(2) Page 3 lines 58-59: “….in plants and animals, …….modified cellular RNAs [6-9].”

The references 6-9 describe not only modifications in plants and animals but also those in microorganisms. Please alter the sentence.

(3) Page 3 lines 74-75: “…. To maintain translational fidelity and efficiency [10, 19, 20].”

As the authors know, the most basic function of m1G37 modification in tRNA is prevention of frameshift error. The reference 24 should be cited in this sentence.

(4) The m1G37 modification may prevent erroneous amino-acylation of tRNAAsp by Arg-RS. See this reference “Perret et al. Nature 344, 787-789 (1990)”.

(5) Please check the citations through the manuscript. For example, Page 4 line 94: the reference [26] may be [27]. Page 4 line 81: Why do the authors cite the reference [27] in this sentence? Comparison of Trm5 and TrmD is summarized in this review “Hori Biomolecules 7, 23 (2017)”.

(6) The abbreviations of tRNAs should be unified. For example, Page 4 line 86: “tRNAPRO and tRNALEU” should be “tRNAPro and tRNALeu”. Figure 4C: “tRNAASP” should be “tRNAAsp”.

(7) Page 5 line 109: the words “Attrm5 plant”. To prevent misunderstanding by the readers, “Attrm5 mutant plant” is recommended.

(8) Page 5 line 117: “Pyrococcus, and Methanococcus”. Please use italic fonts.

(9) Page 9 line 198: “mutant Attrm5 R166D”. Is this a gene name? If so, the words “a catalytic inactive” may be not appropriate.

(10) Page 9 line 212: “T7 polymerase” -> “T7 RNA polymerase”.

(11) Page 10 line 242: “S-adenosyl-methionine (AdoMet)”. These words have already been described in Page 9 line 214.

(12) Page 13 line 313: “KEGG annotation revealed enrichment of proteins involved in photosynthesis, and photosynthetic proteins”. If possible, please add information whether the genes for these proteins are encoded in the nucleus genome. This information may help discussions in Page 17-18.

(13) Page 15 line 353: “suggesting that the loss of m1G37 and m1I37 affects tRNA half stability”. The results in Figure 8 do not show the stability of tRNA half directly. Please revise the sentence.

Page 18 line 438: “….5’ half stability”. Please revise the sentence by the same reason.

(14) Page 15 line 355-364. In general, Ala-RS does not recognize the anticodon-loop. Therefore, the results by the authors seem to be reasonable. Please see this paper “Chong et al. Proc. Natl. Acad. Sci. USA 115, 7527-7532 (2018)”.

(15) Page 16 line 373: “a 60% reduction in m1G and m1I levels”. The reduction in m1I level may be not 60%.

(16) Page 24 line 575: “T7 polymerase” -> “T7 RNA polymerase”

Reviewer #2: The manuscript describes the analysis of the A. thaliana homolog of the TRM5 tRNA methyltransferase, using biochemical and genetic approaches. The data provided strongly support the conclusion that the studied homolog (AtTRM5) is a bonafide m1G37 methyltransferase, by decreased m1G levels in mutant plants, ability of the gene to complement the yeast trm5delta mutant phenotype, and apparent m1G modification activity on a yeast tRNA in the presence of the purified enzyme. These data are all relatively sound (although a few technical issues should be addressed as described below), and therefore are consistent with the expected function of this enzyme. Likewise, the identification of several phenotypes associated with mutant plants are intriguing, and were identified through well-controlled experiments, including complementation by re-expressing the wild-type enzyme in the mutant strains. Again, these data support the conclusion that there is an important function for Trm5 and its methylation activity. Where the paper falls significantly short, however, is in a very limited description of some experiments and methods, making it difficult to assess the data and its significance or quality. Moreover, there is a major gap left unaddressed regarding the possible significance and roles for the other two Trm5 homologs encoded by the A. thaliana genome, as discussed below.

1) Based on the sequence alignment shown in Figure S2, it is unclear why the authors focused on this specific homolog, and not either of the other two identified genes, since conserved motifs are present in all three. Importantly, the sequence of ScTrm5 shown in figure S2B is not complete- seems to start somewhere in the middle of the protein sequence in this alignment (as opposed to S2A which shows the whole thing). The possibilities of redundant or overlapping functions, as has been now identified for many families of enzymes with similar gene duplications in multicellular eukaryotes, can not be ignored. If the authors choose not to look at these other enzymes and their activities through similar experiments, conclusions about the uniqueness or sufficiency of the studied AtTrm5 ortholog should be significantly tempered. For example, the residual m1G (~30%) and m1I (~10%) levels in the mutant plants were dismissed as being the result of modification at position 9, and while this is a possibility, the alternative explanation that these are the result of activity by the other proteins is not considered. The sequence alignments with the other two genes should be moved to the main results for transparency, along with addressing these issues.

2) For the recombinant protein activity assays, it seems that levels of m1G were measured by the MS assay (although not stated in methods where activity assay is described). However, details regarding how "relative nucleoside level" were determined and what this means are not provided. This is significant because if the relative level of 1.0 means 100% modification of G37, then the meaning of the ~1.5X modification at higher levels of enzyme is unclear. Same is true for m1I assay results.

3) The description of data analysis for the deep sequencing results for modification shown in figure 5C are entirely missing, and the data are not clearly presented (greatly oversimplified) because there is no quantification of the results. For this type of approach, it seems that there would have been an analysis of large numbers of reads, so it is unlikely that there was quantitative conversion of the A37 nucleotide to the T or G that is reported in the mutant strains. What were the total numbers of reads, and percentages of T substituted (and other substitutions) vs. G substituted reads from each tRNA? If there was 100% conversion in every read, then this unusual situation (even in other well-studied examples of analyzing tRNA modificaitons by RNAseq there are sequencing errors and some biological signal from unreacted substrates) should be discussed. The sanger results for the trm5-1 mutant (5D) anyway suggest that this is unlikely, since a significant A peak due to (presumably) unreacted A37 is clearly visible in the trace.

4) The location of the GFP reporter fusion (N-or C terminal) is never identified, and description of the construct in the section about fluorescence references the previous section on cloning where GFP is never mentioned. This is relevant because GFP location could affect activity and/or localization of the protein, and certainly if a N-terminal GFP is used, it may not be able to detect mitochondrial signal, if there is one.

5) The description of the newly identified NLS is somewhat confusing- the implication seems to be that the identified importin-dependent NLS is unique to plant Trm5, but yet it is clearly in a highly conserved part fo the enzyme including with ScTrm5?

6) Although it is hard to read figure 7, it is unclear whether figure 7B includes only down-regulated proteins, or all differentially expressed (up and down). If includes both, some presentation of the data that supports the statement that the downregulated ones were specifically over-represented in the photosynthesis related ones is needed.

7) Figure S6, which according to text validates the proteomics data, is missing from the manuscript I received for review. Therefore this claim can not be evaluated.

8) It is hard to evaluate the RNAseq and proteomics results since these are not performed by ribosome profiling (more easy to internally control and standardize). Validation with northern and western for these specific targets identified in Figure S5 (some of the 27, plus some controls) would be needed to confirm. These data are better reserved for a future study in the absence of additional validation or rationalization, especially in light of the inconsistencies described extensiovely in discussion.

9) A control northern experiment for tRNA half levels for a non-substrate tRNA (such as the tRNAArg control) is needed to evaluate the specificity of the observed loss of tRNA half abundance- is this direction due to the loss of trm5 activity, or an indirect effect of the many other phenotypes identified earlier in the paper? Without these data, the conclusions made about this in discussion are somewhat overreaching.

Editorial comments-

1) The introduction needs proofreading and revision for clarity overall, including issues with English grammar/language.

2) Figures are very low quality in the pdf version I got for review- should be addressed in revised version.

6. PLOS authors have the option to publish the peer review history of their article (what does this mean?). If published, this will include your full peer review and any attached files.

Reviewer #1: No

Reviewer #2: No

---

## [Author Response · Author response to Decision Letter 0]

8 Sep 2019

Please find below a point-by-point response to reviewers one and two comments.

Reviewer #1

Response- We thank reviewer one for their very positive recommendation to accept out manuscript for publication after addressing the minor points raised below.

Minor points:

(1) Page 6 lines 139-143. As far as I know, m1I is found only at position 37 in tRNAs from eukaryotes. However, m1I in tRNAs from trm5 mutants is not lost completely. I feel that some explanations for this observation may be necessary.

Response- Edits have been incorporated as suggested, in page 6 line 159-160. The residual m1I in trm5 could be attributed to m1I found at position 57. The reference [33] Torres et al. 2014 “A‐to‐I editing on tRNAs: Biochemical, biological and evolutionary implications” has been added to support the statement.

(2) Page 3 lines 58-59: “….in plants and animals, …….modified cellular RNAs [6-9].” The references 6-9 describe not only modifications in plants and animals but also those in microorganisms. Please alter the sentence.

Response- The sentence has been altered to “ …mRNAs in yeast, plants, and animals” to include microorganisms as described in the references 6-9 and as suggested by the reviewer.

(3) Page 3 lines 74-75: “…. To maintain translational fidelity and efficiency [10, 19, 20].”

As the authors know, the most basic function of m1G37 modification in tRNA is prevention of frameshift error. The reference 24 should be cited in this sentence.

Response- The sentence has been edited to “ …maintain translational fidelity, prevent frameshift error, and efficiency” with the reference (previously reference [24]) [21] Björk GR et al. (1989).” Page 3 line 79. Prevention of translational frameshifting by the modified nucleoside 1-methylguanosine” included as suggested by the reviewer.

(4) The m1G37 modification may prevent erroneous amino-acylation of tRNAAsp by Arg-RS. See this reference “Perret et al. Nature 344, 787-789 (1990)”.

Response- The suggested literature [44] Perret et al. (1990) “Relaxation of a transfer RNA specificity by removal of modified nucleotides” has been included in page 16 lines 404-406 to facilitate the discussion of modification m1G37.

(5) Please check the citations through the manuscript. For example, Page 4 line 94: the reference [26] may be [27]. Page 4 line 81: Why do the authors cite the reference [27] in this sentence? Comparison of Trm5 and TrmD is summarized in this review “Hori Biomolecules 7, 23 (2017)”.

Response- We thank the reviewer for raising this point. The references on page 4 line 87 has been checked and edited, with the reference [28] (previously reference [27]) Noma et al.(2006) “Biosynthesis of wybutosine, a hyper-modified nucleoside in eukaryotic phenylalanine tRNA” removed and replaced by the suggested citation under reference [27] Hori (2017) “Transfer RNA methyltransferases with a SpoU-TrmD (SPOUT) fold and their modified nucleosides in tRNA”.

On page 4 line 99, reference [26] Björk GR et al. (2001) “A primordial tRNA modification required for the evolution of life?” has been removed and was replaced with the reference [28] (previously reference [27]) Noma et al.(2006) “Biosynthesis of wybutosine, a hyper-modified nucleoside in eukaryotic phenylalanine tRNA” as suggested by the reviewer. All references in the manuscript were checked. 

(6) The abbreviations of tRNAs should be unified. For example, Page 4 line 86: “tRNAPRO and tRNALEU” should be “tRNAPro and tRNALeu”. Figure 4C: “tRNAASP” should be “tRNAAsp”.

Response- The abbreviations for tRNA has been edited to ensure that they are unified across the manuscript. This includes the tRNA abbreviations in page 4 line 91 and Figure 4C as mentioned by the reviewer.

(7) Page 5 line 109: the words “Attrm5 plant”. To prevent misunderstanding by the readers, “Attrm5 mutant plant” is recommended.

Response- The suggested edit has been incorporated, for example into page 5 line 116, “Attrm5 plants” has now been changed to “Attrm5 mutant plants”.

(8) Page 5 line 117: “Pyrococcus, and Methanococcus”. Please use italic fonts.

Response- Thank you. This has been accordingly edited. In page 5 line 126 and 127 (previously line 117 page 5), the genus name “Drosophilia, Pyrocococcus, and Methanococcus” has been italicised.

(9) Page 9 line 198: “mutant Attrm5 R166D”. Is this a gene name? If so, the words “a catalytic inactive” may be not appropriate.

Response- Page 9 line 219 (previously line 198), “mutant Attrm5 R166D” refers to the amino acid with mutated amino acid R (Arginine) to D (Aspartic acid). Hence, we retained the words “a catalytic inactive” to describe this loss-of-function protein at the catalytic site. To add clarity, we have also de-italicised Attrm5 which previously caused the confusion.

(10) Page 9 line 212: “T7 polymerase” -> “T7 RNA polymerase”.

Response- “T7 polymerase” has been edited to “T7 RNA polymerase” for page 9 line 233 (previously line 212 of the same page). We have also synchronised the usage of “T7 RNA polymerase” throughout the revised manuscript.

(11) Page 10 line 242: “S-adenosyl-methionine (AdoMet)”. These words have already been described in Page 9 line 214.

Response- The words “S-adenosyl-methionine” in page 10 line 249 (previously line 242) were removed, while the term “S-adenosyl-methionine” will be first mentioned in page 10 line 266.

(12) Page 13 line 313: “KEGG annotation revealed enrichment of proteins involved in photosynthesis, and photosynthetic proteins”. If possible, please add information whether the genes for these proteins are encoded in the nucleus genome. This information may help discussions in Page 17-18.

Response- The genome encoding information for the genes reported in the KEGG analysis has been included in page 13 line 340 and 341“…with most of these proteins encoded by genes found in the nuclear and chloroplast genome.”

(13) Page 15 line 353: “suggesting that the loss of m1G37 and m1I37 affects tRNA half stability”. The results in Figure 8 do not show the stability of tRNA half directly. Please revise the sentence. Page 18 line 438: “….5’ half stability”. Please revise the sentence by the same reason.

Response- Thank you we agree. Page 15 line 380 (previously line 353 of the same page) has been revised and edited to “…suggesting that the loss of m1G37 and m1I37 affects tRNA halve steady state abundance”. We have also revised the sentence in page 19 line 473 (previously page 18 line 438) to “ … tRNA aminoacylation but does affect tRNA-Ala-derived 5’ half steady state abundance”. 

(14) Page 15 line 355-364. In general, Ala-RS does not recognize the anticodon-loop. Therefore, the results by the authors seem to be reasonable. Please see this paper “Chong et al. Proc. Natl. Acad. Sci. USA 115, 7527-7532 (2018)”.

Response- The findings from the suggested paper have been included in page 15-16 line 391 to explain our observations of reduced tRNA-Ala-derived 5’ half steady state abundance, “One explanation to this observation can be attributed to the nature of alanyl-tRNA synthetases that recognizes G3:U70 of tRNA acceptor stem for tRNA charging instead of the anticodon region” , citing reference [41] Chong et al. (2018) “Distinct ways of G:U recognition by conserved tRNA binding motifs.”

(15) Page 16 line 373: “a 60% reduction in m1G and m1I levels”. The reduction in m1I level may be not 60%.

Response- This has been edited. Please refer to page 16 line 413 “With the 60% reduction of m1G and 75% reduction of m1I in trm5 plants”.

(16) Page 24 line 575: “T7 polymerase” -> “T7 RNA polymerase”

Response- “T7 polymerase” has been edited to “T7 RNA polymerase” throughout the revised manuscript.

Reviewer #2: 

Overall Response- We thank reviewer two for their largely positive review and address their comments below. We agree that some technical information was not adequately described or were described elsewhere in the materials and methods. We have now addressed these points. While reviewer 2 suggested one additional experiment, we note the Editor clearly stated that no additional experiments were required before publication.

1) Based on the sequence alignment shown in Figure S2, it is unclear why the authors focused on this specific homolog, and not either of the other two identified genes, since conserved motifs are present in all three. Importantly, the sequence of ScTrm5 shown in figure S2B is not complete- seems to start somewhere in the middle of the protein sequence in this alignment (as opposed to S2A which shows the whole thing). The possibilities of redundant or overlapping functions, as has been now identified for many families of enzymes with similar gene duplications in multicellular eukaryotes, can not be ignored. If the authors choose not to look at these other enzymes and their activities through similar experiments, conclusions about the uniqueness or sufficiency of the studied AtTrm5 ortholog should be significantly tempered. For example, the residual m1G (~30%) and m1I (~10%) levels in the mutant plants were dismissed as being the result of modification at position 9, and while this is a possibility, the alternative explanation that these are the result of activity by the other proteins is not considered. The sequence alignments with the other two genes should be moved to the main results for transparency, along with addressing these issues.

Response- We have added a sentence, page 5 lines 133 and 134, that addresses why experiments focussed on AtTRM5. Also on page 6, lines 151-153, we acknowledge that some observations may be explained by AtTRM5B or AtTRM5C.

2) For the recombinant protein activity assays, it seems that levels of m1G were measured by the MS assay (although not stated in methods where activity assay is described). However, details regarding how "relative nucleoside level" were determined and what this means are not provided. This is significant because if the relative level of 1.0 means 100% modification of G37, then the meaning of the ~1.5X modification at higher levels of enzyme is unclear. Same is true for m1I assay results.

Response- Thank you reviewer two. The materials and methods section were updated to include more technical information, page 6 lines 679-690. The y axis of figures 4 and 5 were updated.

3) The description of data analysis for the deep sequencing results for modification shown in figure 5C are entirely missing, and the data are not clearly presented (greatly oversimplified) because there is no quantification of the results. For this type of approach, it seems that there would have been an analysis of large numbers of reads, so it is unlikely that there was quantitative conversion of the A37 nucleotide to the T or G that is reported in the mutant strains. What were the total numbers of reads, and percentages of T substituted (and other substitutions) vs. G substituted reads from each tRNA? If there was 100% conversion in every read, then this unusual situation (even in other well-studied examples of analyzing tRNA modificaitons by RNAseq there are sequencing errors and some biological signal from unreacted substrates) should be discussed. The sanger results for the trm5-1 mutant (5D) anyway suggest that this is unlikely, since a significant A peak due to (presumably) unreacted A37 is clearly visible in the trace.

Response- The materials and methods sub-section heading was edited to highlight the relevant section, page 25 line 625. Based on the reads of the GATK results (which the SNPs were validated to be significant), A -> G conversions were consistently observed in trm5 mutants, with no A-> T conversions in the data. As for the WT data, there is a mix of A-> and A->G conversion, which is why the data has now been presented in a qualitative format. Figure 5 legend, page 37 line 1044 was edited to highlight the qualitative nature of the assay. Instead, a dichotomous system is adapted to explain the observation. As mentioned in our manuscript, despite base conversion can be used to determine the possibility of modification change on RNA sequences, it is also known that RNA modification stalls reverse transcriptase activity, hence highlighting the technical difficulty in obtaining high read coverage for quantitative analysis as we hoped for.

4) The location of the GFP reporter fusion (N-or C terminal) is never identified, and description of the construct in the section about fluorescence references the previous section on cloning where GFP is never mentioned. This is relevant because GFP location could affect activity and/or localization of the protein, and certainly if a N-terminal GFP is used, it may not be able to detect mitochondrial signal, if there is one.

Response- The materials and methods section, page 21 lines 535-540, was edited to show that the GFP was a C-terminal translational fusion to TRM5. 

5) The description of the newly identified NLS is somewhat confusing- the implication seems to be that the identified importin-dependent NLS is unique to plant Trm5, but yet it is clearly in a highly conserved part fo the enzyme including with ScTrm5?

Response- This has now been addressed in the edited manuscript. According to Björk et al. 2001, EMBO J (2001)20:231-239, no known CLS or bipartite NLS signal was known from Trm5p. Our multiple sequence alignment results between Arabidopsis thaliana Trm5 homologues, especially AtTRM5 where the alpha-importin dependent NLS sequence is detected, and ScTrm5p showed only several amino acids were conserved in the region.

6) Although it is hard to read figure 7, it is unclear whether figure 7B includes only down-regulated proteins, or all differentially expressed (up and down). If includes both, some presentation of the data that supports the statement that the downregulated ones were specifically over-represented in the photosynthesis related ones is needed.

Response- We agree that the resolution of Figure 7 was decreased in the PLoS PDF conversion system. The original uploaded imagine was a clear high-resolution figure. We disagree with the reviewer that the figure represents the downregulated photosynthesis proteins only. The GO and KEGG analysis takes into account both upregulated and downregulated proteins. 

7) Figure S6, which according to text validates the proteomics data, is missing from the manuscript I received for review. Therefore, this claim cannot be evaluated.

Response- We apologise for this error. Figure S6 is included in the resubmission.

8) It is hard to evaluate the RNAseq and proteomics results since these are not performed by ribosome profiling (more easy to internally control and standardize). Validation with northern and western for these specific targets identified in Figure S5 (some of the 27, plus some controls) would be needed to confirm. These data are better reserved for a future study in the absence of additional validation or rationalization, especially in light of the inconsistencies described extensively in discussion.

Response- Some of these candidates are validated in Figure S6 western blot results (which we apologize for missing out this data during the previous submission). Our western blot results are consistent with our findings reported in the RNA-seq/Proteomics data.

9) A control northern experiment for tRNA half levels for a non-substrate tRNA (such as the tRNAArg control) is needed to evaluate the specificity of the observed loss of tRNA half abundance- is this direction due to the loss of trm5 activity, or an indirect effect of the many other phenotypes identified earlier in the paper? Without these data, the conclusions made about this in discussion are somewhat overreaching.

Response- While it is possible that our result may be indirect or non-specific, we do clearly state the result “suggesting that the loss of m1G37 and m1I37 affects tRNA halve steady state abundance.”, page 15 lines 381 and 382. We thank the Editor for clearly stating that no additional experiments were required before publication.

Editorial comments-

1) The introduction needs proofreading and revision for clarity overall, including issues with English grammar/language.

Response- A number of edits have been made throughout the manuscript. Please see the “marked up” version of the manuscript. Examples includes page 2, lines 48 and 49.

2) Figures are very low quality in the pdf version I got for review- should be addressed in revised version.

Response- High-quality figures were uploaded to the PLoS system but unexpectedly the PDF version of the combined figures and text had some low-resolution figures.

Data availability is clearly shown and available to the community- see page 34 line 976.

---

## [Decision Letter · Decision Letter 1]

14 Oct 2019

PONE-D-19-17540R1

Arabidopsis TRM5 Encodes a Nuclear-localised Bifunctional tRNA Guanine and Inosine-N1-methyltransferase that is Important for Growth

PLOS ONE

Dear Dr Searle,

Thank you for submitting your manuscript to PLOS ONE. After careful consideration, we feel that it has merit but does not fully meet PLOS ONE’s publication criteria as it currently stands. Therefore, we invite you to submit a revised version of the manuscript that addresses the points raised during the review process.

Your manuscript has been evaluated by the same two reviewers as previously. Although Reviewer #1 is now fully satisfied by your revised manuscript, Reviewer #2 has expressed some concerns, particularly on the quality of your validation data (which were not available for review in the previous version). I have also read the manuscript myself and I would like to invite you to submit a new version addressing the points raised by Reviewer #2 and myself.

As the link between Figure S6 and the proteomic data remains somewhat unclear (see the reviewer's comments below), I would suggest removing this figure from the manuscript, provided that you clearly state elsewhere in the manuscript that the results have not been subjected to validation by an alternative method. 

I am also asking you to supply additional information on some aspects of Supplementary Table S3, Please clarify how the p-values in the table have been adjusted (i.e. corrected for multiple testing), as this information is not included in the Methods section. Please note that if they have not been corrected, the validity of some of your conclusions might be compromised. In this table, please also include the raw values for each genotype (i.e. not only their ratios).

Please see below some additional minor comments. Note that PLOS ONE does not copyedit accepted articles, and all problems in the text must be fixed before acceptance.

We would appreciate receiving your revised manuscript by Nov 28 2019 11:59PM. To enhance the reproducibility of your results, we recommend that if applicable you deposit your laboratory protocols in protocols.io, where a protocol can be assigned its own identifier (DOI) such that it can be cited independently in the future. For instructions see: http://journals.plos.org/plosone/s/submission-guidelines#loc-laboratory-protocols

We look forward to receiving your revised manuscript.

Kind regards,

Hector Candela, Ph.D.

Academic Editor

PLOS ONE

Additional Editor Comments (if provided):

Line 47: Split the two sentences "...mutants. However, ..."

l. 85: remove comma before "catalyses"

l. 89: add a comma after "TrmD".

l. 91: have -> has

Line 150-151: "in Arabidopsis thaliana" is said twice

Line 186: Use uppercase characters for the gene name. Regulators -> REGULATOR (no S)

Line 198: something is wrong in this sentence ("a yeast")

Reviewers' comments:

Reviewer's Responses to Questions

**Comments to the Author**

1. If the authors have adequately addressed your comments raised in a previous round of review and you feel that this manuscript is now acceptable for publication, you may indicate that here to bypass the “Comments to the Author” section, enter your conflict of interest statement in the “Confidential to Editor” section, and submit your "Accept" recommendation.

Reviewer #1: (No Response)

Reviewer #2: (No Response)

2. Is the manuscript technically sound, and do the data support the conclusions?

Reviewer #1: Yes

Reviewer #2: No

3. Has the statistical analysis been performed appropriately and rigorously? 

Reviewer #1: Yes

Reviewer #2: No

4. Have the authors made all data underlying the findings in their manuscript fully available?

Reviewer #1: Yes

Reviewer #2: Yes

5. Is the manuscript presented in an intelligible fashion and written in standard English?

Reviewer #1: Yes

Reviewer #2: Yes

6. Review Comments to the Author

Reviewer #1: The authors adequately addressed my concerns. I would like to suggest the accept of this revised manuscript.

Reviewer #2: Revisions have been made in response to several issues raised previously. The presentation of the deep sequencing analysis for tRNA modification at position 37 is somewhat improved here, and the qualitative description of the results is ok, but as described, it is unclear exactly what these data add above what is clearly visible already from the sanger sequencing result in panel D. nonetheless, I do not have technical issues with the way it is presented now. However, the addition of the missing data from the last round has raised new questions that need to be addressed because they go directly to the issue of confidence in the interpretation that photsynthetic gene expression is affected specifically by loss of trm5.

1) Now that western validation data are available, the interpretation that the results were consistent with the proteomics needs to be supported with much more analysis. I see no names of proteins identified in Figure 7 (call the dots out on the proteomics that were tested in the western, as is commonly done) or discussed in the text so that I can evaluate how the western data do, or do not, support the proteomics by a direct comparision.

- major issue: there is no quantification provided in Figure S6- I see 6 proteins, with some whose levels seem to change to some extent relative to WT and some that don't appear to change as much, but of course this would require actual quantification in some way. The problem is normalization for loading, which the authors suggest is supposed to be done visually by the coomassie staining, although this is of course not appropriate at all because we are looking at one protein (which one?) and it is not the typical way to normalize by a protein western control signal for a protein that should not be changing. However, even trying to interpret this result (by eye) seems to clearly show heavier bands in the coomassie WT "100%" lane than in the corresponding trm5-1 lane. As presented, these data definitely do not meet typical standards for scientific rigor.

- were some of these proteins controls that are used to show that levels did not change, as expected based on the proteomics?

- I assume the 100% listed for trm5-1 indicates some fraction of the sample loaded, but this doesn't quite make sense- this is better phrased with some indication of exact total protein loaded per lane? This is a minor point, but would help improve clarity.

Bottom line, the superficial way the validation is performed and described significantly impacts the confidence in the proteomics data, which are a major conclusion of the paper.

2) a related issue is with the RNAseq data used to suggest targets whose regulation is at the level of translation. Since validation of the RNAseq data are not provided (which is the standard in the field), and because now it is clear that the western validation (now shown) did not focus specifically on any of these proteins, and the quantitative data needed to rigorosuly support the validity of the proteomics are also absent (see point above), this result identifying some proteins whose may be regulated at the level of translation can not be viewed as well-supported by any current standards in the field. This highly speculative result could be removed to discussion, or quite honestly from the paper entirely, without further substantiation.

In short, this paper nicely demonstrates biochemical and biological functions for a homolog of the tRNA methyltransferase trm5 in A. thaliana. Results directed toward that goal ,and characterization of mutant plants, which already suggests interesting things to be followed up on, are a strength and are rigorously described. Beyond that, the attempts to delve deeper into underlying changes in gene expression are not well-supported by the data presented, which in many cases do not meet even the bare minimum of standards for critical analysis for these kinds of studies.

7. PLOS authors have the option to publish the peer review history of their article (what does this mean?). If published, this will include your full peer review and any attached files.

Reviewer #1: No

Reviewer #2: No

---

## [Author Response · Author response to Decision Letter 1]

25 Oct 2019

Please find below a point-by-point response to the editor, reviewers one and two comments. 

Reponses are in blue coloured font.

Editor’s comments:

(1) Comment- As the link between Figure S6 and the proteomic data remains somewhat unclear (see the reviewer's comments below), I would suggest removing this figure from the manuscript, provided that you clearly state elsewhere in the manuscript that the results have not been subjected to validation by an alternative method. 

Response -The supplementary figure S6, along with the figure captions and all previous mentions in the manuscript have been removed. We have also added a senetence that in the future that validation of the proteomics data will be required via quantitative western blotting to verify the direct or indirect effect of the loss of TRM5. Please refer to page 14 lines 324-327 of the manuscript for the edits.

(2) Comment- I am also asking you to supply additional information on some aspects of Supplementary Table S3, Please clarify how the p-values in the table have been adjusted (i.e. corrected for multiple testing), as this information is not included in the Methods section. Please note that if they have not been corrected, the validity of some of your conclusions might be compromised. In this table, please also include the raw values for each genotype (i.e. not only their ratios).

Response - Additional information on the proteomics data, including the raw values of each genotype have been included in the updated Supplementary Table 3. The p-values have been corrected using Benjamini-Hochberg (BH) correction method and the methods section in the manuscript has been updated accordingly. The BH corrected p-values have been included in supplementary Table 3. 

Additional Editor Comments (if provided):

(3) Comment- Line 47: Split the two sentences "...mutants. However, ..."

Response - The suggested edits were incorporated. The sentences are split into two at page 2 line 47-48. 

(4) Comment- l. 85: remove comma before "catalyses"

Response -The sentence has been amended with the comma removed. Please refer to page 4 line 86.

(5) Comment- l. 89: add a comma after "TrmD".

Response - A comma has been added after TrmD. Please refer to page 4 line 89.

(6) Comment- l. 91: have -> has

Response - This sentence was corrected as suggested. Please refer to page 4 line 91.

(7) Comment- Line 150-151: "in Arabidopsis thaliana" is said twice

Response - The redundant “in Arabidopsis thaliana” mentioned in the manuscript was removed as suggested. Please refer to page 6 line 151.

(8) Comment- Line 186: Use uppercase characters for the gene name. Regulators -> REGULATOR (no S)

Response - The gene name PSEUDO RESPONSE REGULATOR 7 has been corrected to all upper-case characters at page 8 line 190.

(9) Comment- Line 198: something is wrong in this sentence ("a yeast")

Response - The sentence was amended by removing “..a yeast”. Please refer to page 8 line 201.

(10) We proofread our manuscript prior to resubmission. 

With this resubmission, we attached the corrected Fig 1, which in Fig 1 panel A and B the misspelled “SITRM5” has been corrected to “SiTRM5”.

Reviewer #1:

We thank reviewer 1 for the positive feedback on the manuscript revisions.

Reviewer #2: 

(11) Now that western validation data are available, the interpretation that the results were consistent with the proteomics needs to be supported with much more analysis. I see no names of proteins identified in Figure 7 (call the dots out on the proteomics that were tested in the western, as is commonly done) or discussed in the text so that I can evaluate how the western data do, or do not, support the proteomics by a direct comparison.

- major issue: there is no quantification provided in Figure S6- I see 6 proteins, with some whose levels seem to change to some extent relative to WT and some that don't appear to change as much, but of course this would require actual quantification in some way. The problem is normalization for loading, which the authors suggest is supposed to be done visually by the coomassie staining, although this is of course not appropriate at all because we are looking at one protein (which one?) and it is not the typical way to normalize by a protein western control signal for a protein that should not be changing. However, even trying to interpret this result (by eye) seems to clearly show heavier bands in the coomassie WT "100%" lane than in the corresponding trm5-1 lane. As presented, these data definitely do not meet typical standards for scientific rigor.

- were some of these proteins controls that are used to show that levels did not change, as expected based on the proteomics?

- I assume the 100% listed for trm5-1 indicates some fraction of the sample loaded, but this doesn't quite make sense- this is better phrased with some indication of exact total protein loaded per lane? This is a minor point, but would help improve clarity.

Bottom line, the superficial way the validation is performed and described significantly impacts the confidence in the proteomics data, which are a major conclusion of the paper.

Response - The supplementary figure S6 has been removed from the manuscript after careful revision of this manuscript based on the reviewer #2’s comments.

(12) a related issue is with the RNAseq data used to suggest targets whose regulation is at the level of translation. Since validation of the RNAseq data are not provided (which is the standard in the field), and because now it is clear that the western validation (now shown) did not focus specifically on any of these proteins, and the quantitative data needed to rigorosuly support the validity of the proteomics are also absent (see point above), this result identifying some proteins whose may be regulated at the level of translation can not be viewed as well-supported by any current standards in the field. This highly speculative result could be removed to discussion, or quite honestly from the paper entirely, without further substantiation.

Response - We have edited the manuscript and have soften our results section according to the reviewer’s suggestion. Please refer to page 14 lines 324-327 of the manuscript for the edits.

We thank reviewer 2 for the constructive feedback on the manuscript.

---

## [Editor Report · Decision Letter 2]

29 Oct 2019

Arabidopsis TRM5 Encodes a Nuclear-localised Bifunctional tRNA Guanine and Inosine-N1-methyltransferase that is Important for Growth

PONE-D-19-17540R2

Dear Dr. Searle,

We are pleased to inform you that your manuscript has been judged scientifically suitable for publication and will be formally accepted for publication once it complies with all outstanding technical requirements.

With kind regards,

Hector Candela, Ph.D.

Academic Editor

PLOS ONE
---

## [Editor Report · Acceptance letter]

7 Nov 2019

PONE-D-19-17540R2 

Arabidopsis TRM5 Encodes a Nuclear-localised Bifunctional tRNA Guanine and Inosine-N1-methyltransferase that is Important for Growth 

Dear Dr. Searle:

I am pleased to inform you that your manuscript has been deemed suitable for publication in PLOS ONE. Congratulations! Your manuscript is now with our production department. 

With kind regards,

on behalf of

Dr. Hector Candela 

Academic Editor

PLOS ONE